# A quantitative model for human neurovascular coupling with translated mechanisms from animals

**Sebastian Sten**[1,2,3☯¤], **Henrik Podéus**[3☯], **Nicolas Sundqvist**[2,3], **Fredrik Elinder**[4], **Maria Engström**[1,2☯], **Gunnar Cedersund**[2,3☯]*

**1** Department of Health, Medicine and Caring Sciences, Linköping University, Linköping, Sweden, **2** Center for Medical Image Science and Visualization (CMIV), Linköping University, Linköping, Sweden, **3** Department of Biomedical Engineering, Linköping University, Linköping, Sweden, **4** Department of Biomedical and Clinical Sciences, Linköping University, Linköping, Sweden

☯ These authors contributed equally to this work.
¤ Current address: Drug Metabolism and Pharmacokinetics, Research and Early Development, Cardiovascular, Renal and Metabolism (CVRM), BioPharmaceuticals R&D, AstraZeneca, Gothenburg, Sweden
* gunnar.cedersund@liu.se

**Data Availability Statement:** No new data is presented in this work. The code necessary for reproducing all the figures and results reported herein is provided as a public GitHub repository as

## Abstract

Neurons regulate the activity of blood vessels through the neurovascular coupling (NVC). A detailed understanding of the NVC is critical for understanding data from functional imaging techniques of the brain. Many aspects of the NVC have been studied both experimentally and using mathematical models; various combinations of blood volume and flow, local field potential (LFP), hemoglobin level, blood oxygenation level-dependent response (BOLD), and optogenetics have been measured and modeled in rodents, primates, or humans. However, these data have not been brought together into a unified quantitative model. We now present a mathematical model that describes all such data types and that preserves mechanistic behaviors between experiments. For instance, from modeling of optogenetics and microscopy data in mice, we learn cell-specific contributions; the first rapid dilation in the vascular response is caused by NO-interneurons, the main part of the dilation during longer stimuli is caused by pyramidal neurons, and the post-peak undershoot is caused by NPY-interneurons. These insights are translated and preserved in all subsequent analyses, together with other insights regarding hemoglobin dynamics and the LFP/BOLD-interplay, obtained from other experiments on rodents and primates. The model can predict independent validation-data not used for training. By bringing together data with complementary information from different species, we both understand each dataset better, and have a basis for a new type of integrative analysis of human data.

## Author summary

The neurovascular coupling (NVC) is the basis for functional magnetic resonance imaging (fMRI), since the NVC connects neural activity with the observed hemodynamic

MATLAB scripts and is available from https://github.com/Podde1/A-quantitative-model-for-human-neurovascular-coupling-with-translated-mechanisms-from-animals.git.

**Funding:** This work was supported by the Swedish Research Council (Grant IDs: 2018-05418 and 2018-03319, G.C.; 2018-03391, ME. https://www.vr.se/english.html). Additional support for GC came from CENIIT, Center for Industrial Information Technology, (ID:15.09. http://ceniit.lith.liu.se/), the Swedish Foundation for Strategic Research (ID: ITM17-0245, https://strategiska.se/en/), SciLifeLab National COVID-19 Research Program, financed by the Knut and Alice Wallenberg Foundation (ID: 2020.0182, https://www.scilifelab.se/pandemic-response/covid-19-research-program/), the H2020 project PRECISE4Q, Personalised Medicine by Predictive Modelling in Stroke for better Quality of Life, (ID: 777107, https://precise4q.eu/), the Swedish Fund for Research without Animal Experiments (ID: F2019-0010, https://forskautandjurforsok.se/swedish-fund-for-research-without-animal-experiments/), ELLIIT, Excellence Center at Linköping – Lund in Information Technology, (ID: 2020-A12, https://elliit.se/), VINNOVA (VisualSweden) and VINNOVA together with MedTech4Health and SweLife (ID: 2020-04711, https://www.vinnova.se/en/). Additional funding for ME came from the Swedish Brain Foundation (https://www.hjarnfonden.se/). The funders had no role in study design, data collection and analysis, decision to publish, or preparation of the manuscript.

**Competing interests:** The authors have no competing interests to declare.

changes. This connection is highly complex, which warrants a model-based analysis. However, even though NVC-data from several species and many relevant variables are available, a mathematical model for all these data is still missing. Herein, we combine experimental data from mice, monkeys, and humans, to develop a comprehensive model for NVC. Importantly, our new approach to modelling propagates the qualitative insights from each species to the subsequent analysis of data from other species. In mice, we unravel the role of different neuronal sub-populations when producing a biphasic response to prolonged sensory stimulations. The qualitative role of these sub-populations is preserved when analysing primate data. These primate data add knowledge on the interplay between local field potential (LFP) and vascular changes. Similarly, these pre-clinical qualitative insights are propagated to analysis of human data, which contain additional insights regarding blood flow and volume in arterioles and venules, during both positive and negative responses. This work illustrates how data with complementary information from different species can be combined, so that qualitative insights from animals are preserved in the quantitative analysis of human data.

## 1. Introduction

The adult brain constitutes only ~2% of the total bodyweight of an average adult, but accounts for ~20% of a body's total energy consumption [1]. To meet the high energy demand, the brain requires a continuous supply of metabolic substrates such as glucose and oxygen [2]. This continuous supply of substrates is met by diffusion and receptor-mediated transport from the blood to cerebral tissue, through the capillaries. Thus, as first reported by Roy and Sherrington in 1890 [3], there is a tight temporal and spatial connection between brain activity and hemodynamics. This connection is commonly referred to as the neurovascular coupling (NVC). The NVC describes the coupling between neural cells and blood vessels and involves changes in the cerebral blood flow (CBF), cerebral blood volume (CBV), and the cerebral metabolic rate of oxygen ($CMRO_2$). The NVC is mediated by the synaptic activity of neurons–brain activity–that trigger the release of different vasoactive molecules [4–8] (see Fig 1A for an overview of central signaling pathways). These molecules induce changes in CBF and CBV by acting on vascular smooth muscle cells and pericytes, which enwrap arterioles and capillaries, respectively. Detailed knowledge about NVC is important because it underpins neuroimaging techniques such as functional magnetic resonance imaging (fMRI). These techniques are based on the assumption that hemodynamic changes are a proxy for neural activity [9–11]. However, the interpretation of fMRI data is usually done in a purely statistical way that ignores the underlying biological mechanisms. Thus, a quantitative and mechanistically resolved model for the NVC is missing.

 Previous studies of the NVC have employed a variety of different measurement strategies, each with its possibilities, strengths, and drawbacks [12] (Fig 1B). One technique is the usage of *in vitro* experiments where brain slices are studied to elucidate intracellular signaling intermediaries in different cell types ([5,6], and references therein). While this technique allows for detailed probing of intracellular mechanisms, it does not allow for probing the interaction between these cell types and blood flow. Another technique is invasive *in vivo* animal experiments. Such experiments are predominantly carried out during anesthesia in rodents [13,14]. While such experiments have limited capability to measure intracellular signaling intermediaries, they allow for examinations of the crosstalk between blood supply and brain activity, and measurements of a wide variety of entities, such as CBF, CBV, vessel diameter,

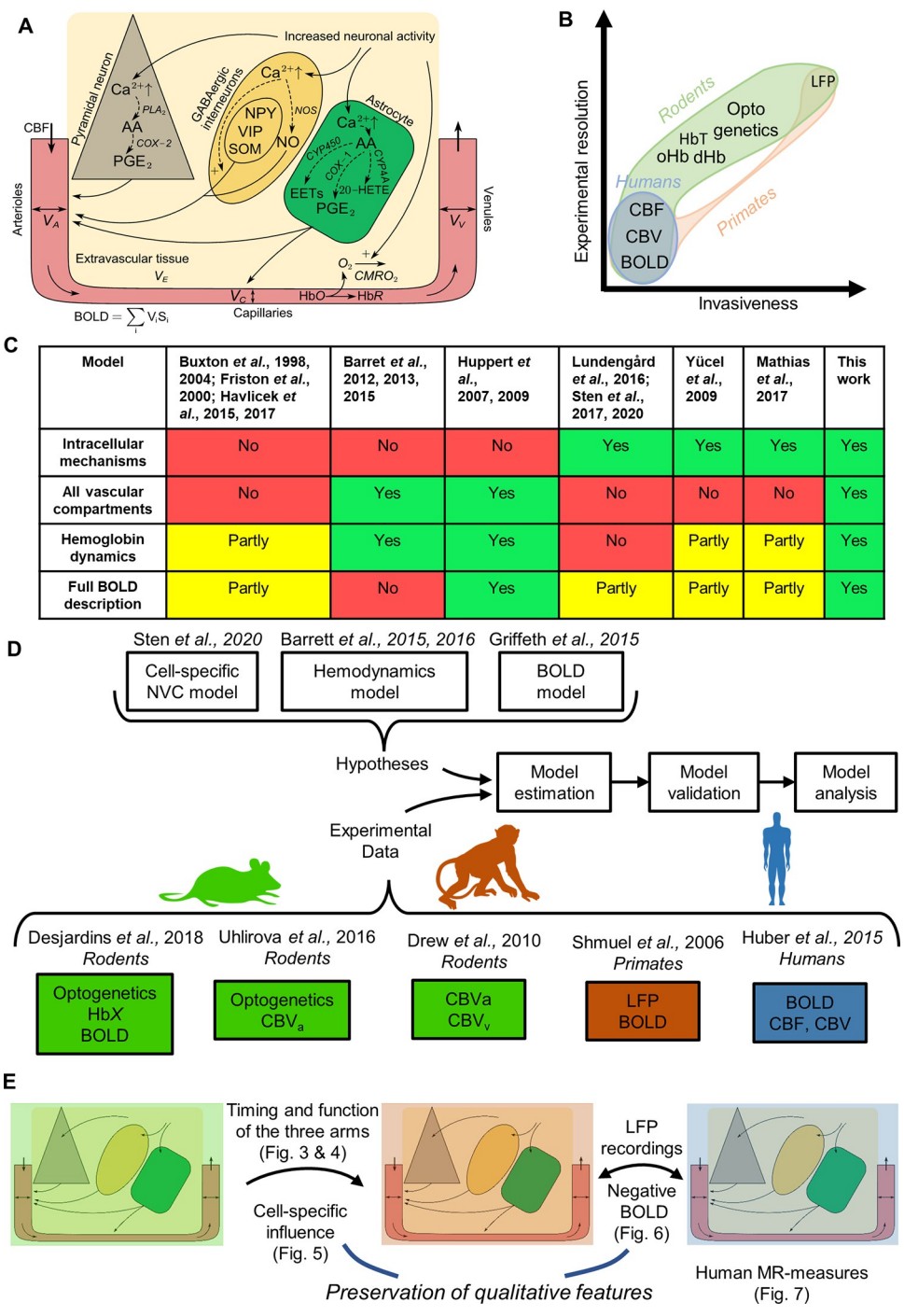

**Fig 1. Overview of the presented study.** A: Simplified schematic illustration of the cellular pathways underlying the NVC. Neuronal signaling activates GABAergic interneurons, pyramidal neurons, and astrocytes by stimulating the influx of $Ca^{2+}$. $Ca^{2+}$ facilitates signaling pathways in the different cells. In GABAergic interneurons, $Ca^{2+}$ promotes nitric oxide (NO) through the upregulation of nitric oxide synthase (NOS). Furthermore, $Ca^{2+}$ also facilitates the release of different neuropeptides: neuropeptide Y (NPY), vasoactive intestinal peptide (VIP), and somatostatin (SOM). In pyramidal neurons, $Ca^{2+}$ promotes the synthesis of arachidonic acid (AA) via phospholipase A2 ($PLA_2$), which in turn is metabolized to prostaglandin E2 ($PGE_2$) via cyclooxygenase-2 (COX-2). In astrocytes, arachidonic acid is synthesized via phospholipase D2 ($PLD_2$) and subsequently metabolized into three different vasoactive molecules: $PGE_2$ via cyclooxygenase-1 (COX-1), epoxyeicosatrienoic acid (EET) via cytochrome P450 (CYP) epoxygenase, and 20-Hydroxyeicosatetraenoic acid (20-HETE) via CYP4A. Together these vasoactive messengers act on arterioles and capillaries to modulate the vessel diameters, where the neuronal messengers act primarily on the

arterioles, while the astrocytes act on both arterioles and capillaries. B: Overview of commonly used experimental techniques in rodents, primates, and humans, depicted along two axes: the spatial and temporal resolution of what is measured, and how invasive the technique is. C: Comparison of different published models describing the neurovascular coupling (NVC) with regards to different mechanisms. The notations used in the table are: 'Yes' on a green background if the model features the stated mechanism, 'Partly' on a yellow background if the model has a description that is not fully satisfying, and 'No' on a read background if there is no description of the mechanism. As seen, no model exists that can describe every mechanism. D: Overview of the study. We have collected already published experimental data from different species and/or experimental techniques and combined these experimental data with the development of a mathematical model, which builds upon three already published models. E: Preservation of qualitative features between the three species, including an overview of specific features and in which figures these features are visualized.

hemoglobin, and oxygen saturation changes, as well as electrical activity such as local field potential (LFP) [13–15]. A key possibility in such animal experiments is the usage of optogenetic techniques [16], which allow for the activation of specific neuronal cell types. While most such animal experiments are done in rodents, Logothetis *et al. (2001)*, have done important studies in higher primates [9,10,17,18]. In two of these studies, they measured LFP and fMRI simultaneously [9,10]. Most of these animal studies are performed with the use of anesthetics, which affects neuronal excitability, cerebral metabolism, and vascular reactivity amongst other physiological processes [19–22], which complicates the interpretation of such studies. Finally, a variety of non-invasive techniques are also available in humans, most of which use magnetic resonance imaging (MRI). Using MRI, one can measure blood oxygen level-dependent (BOLD) responses to different stimuli, and CBF and CBV. In summary, components of the NVC can be measured by different techniques in different experimental systems, but no system alone allows for simultaneous measurement of all these components (Fig 1B) in any species.

The NVC is highly conserved across different species, but despite this, there does not exist a systematic overview or synthesis of different data sources. The NVC is often non-invasively characterized using the canonical hemodynamic response function (HRF) extracted from BOLD-fMRI. The HRF consists of two or three different phases: a debated initial dip, the main response, and a post-peak undershoot. This qualitative shape of the HRF is consistent across species, such as rodents, primates, and humans. Furthermore, also the timing is preserved across these species: in response to a sub-second stimulus, the peak of the main response lies at approximately three to six seconds after a stimulus, and the entire HRF usually lasts 15–20 seconds [23–25]. Therefore, it is reasonable to assume that the mechanisms generating the NVC in the different species are highly preserved. Nevertheless, quantitative details will differ between species, and even between different subjects and experimental settings within the same species. One way to deal with these types of complex data-analysis situations is to use mathematical modeling.

By mathematical modeling, various aspects of the NVC have previously been modeled, even though no cross-species model has been presented. A summary of previous NVC models, and their strengths and weaknesses, is found in Fig 1C. In the standard interpretation of BOLD-fMRI, activity is equated by correlation with the canonical HRF ([26] and references therein). This approach can provide an estimate of where, in the brain, neuronal activity is present, but this approach ignores the neural and vascular mechanisms involved in the NVC. One of the first descriptions of these mechanisms was made with the 'Balloon' model [27–29] (Fig 1C, model column 1). In this model, the volume of the venules is described by similar mechanisms as those governing the expansion of a balloon. While the Balloon model describes a reasonable series of steps involving neuronal activity, CBF, CBV, CMRO$_2$, and the BOLD signal, the equations describing each of these individual steps are purely phenomenological. The

most mechanistically correct way to describe blood flow in vessels is to solve the Navier-Stokes equations [30,31]. In practice, a simplified approach called Windkessel models using lumped parameter models and ordinary differential equations is adequate to provide a sufficient description of the interplay between CBF and CBV [32,33]. There are a few models that describe the interplay between Windkessel dynamics in arterioles and venules, and their interplay with the BOLD response [28,34–37] (Fig 1C, model column 1–3). However, no such model describes the crosstalk between these processes and the intracellular signaling resulting in the release of vasoactive substances (Fig 1A and 1C). Finally, a few models describe these intracellular signaling cascades and their impact on the blood flow regulation, even though these models use a single blood compartment [38–41] (Fig 1C, model column 4–6). One of the more advanced models describing intracellular pathways is based on optogenetics data, which has unraveled the crosstalk between pyramidal cells and two different types of interneurons [42] (Fig 1C, column 4). However, no model of the NVC describes such intracellular pathways together with all the other above-described data available in rodents, primates, and humans.

Herein, we present a first mathematical model that covers multiple aspects of the NVC into one complete model (Figs 1C–1E and 2A–2C). We have connected a mechanistic NVC model for the control of arterioles [42] with a Windkessel model of blood flow, pressure, volume, and hemoglobin content in arterioles, capillaries, and venules [34,43,44], which finally is combined into a complete description of the BOLD signal using the model by Griffeth and Buxton [45] (Figs 1D and 2). Our approach is the first model that can describe previously published data from different species (humans, mice, and chimpanzees), consisting of multiple different measurement variables (diameter change of the vessels, different measures of CBV, CBF, transient hemoglobin changes, BOLD responses, and LFP) gathered using both optogenetic and sensory perturbations. Using a novel approach to preserve qualitative features, the model both draws clear insights from each dataset–in rodents (Figs 3–5), primates (Fig 6), and humans (Fig 7)–and preserves those insights when interpreting the other datasets (Fig 1E). Furthermore, the model can successfully predict independent experimental data used for validation (Figs 1D, 5J, 7B, 7C and 7E). To the best of our knowledge, this model constitutes the most complete mathematical functional description of the NVC to date.

## 2. Results

We have combined three models (Fig 1D), describing different aspects of the NVC into one combined model (Fig 2). The combined model is fitted to data from each of the five studies included herein: i) Drew *et al.* [46] (Section 2.1.1, Figs 3 and 4), ii) Uhlirova *et al.* [47] (Section 2.1.3, S4 Fig), iii) Desjardins *et al.* [48] (Section 2.1.4, Fig 5), iv) Shmuel *et al.* [10] (Section 2.2, Fig 6), and v) Huber *et al.* [49] (Section 2.3, Fig 7). For all these data, simulations are shown together with the data, and for all these data the model passes a $\chi^2$-test. We also saved some data for validation (Figs 1D, 5J, 7B, 7C and 7E). From each dataset, new mechanistic insights are obtained, which are transferred to the analysis of the next dataset. More specifically, in Figs 5G–5I, 6C–6F and 7F–7K, there is an incrementally growing list of plots, which show preserved and translated mechanisms (Fig 1E).

### 2.1 NVC measures in awake and anesthetized mice

**2.1.1 Fractional diameter change of arterioles and venules in awake mice.**   The model was trained using experimental data extracted from the study of Drew *et al.* [46], consisting of arteriolar (Fig 3A–3C, red symbols) and venular (Fig 3D–3F, blue symbols) diameter changes induced by a sensory whisker stimulation for three different durations: 20 ms (Fig 3A and 3D), 10 s (Fig 3B and 3E) and 30 s (Fig 3C and 3F). The stimulation consisted of air puffs (8 Hz, 20

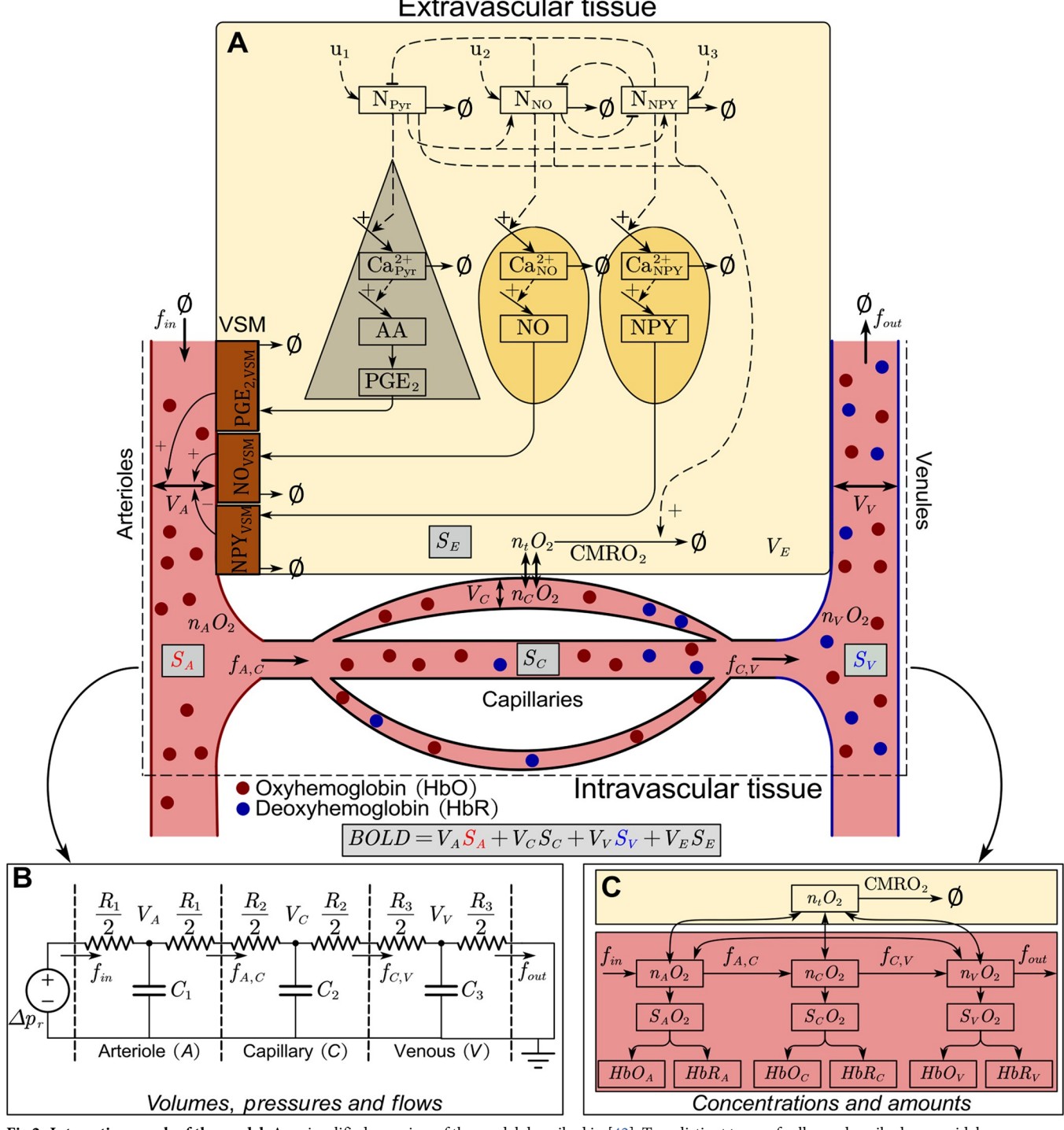

**Fig 2. Interaction graph of the model.** A: a simplified overview of the model described in [42]. Two distinct types of cells are described: pyramidal neurons (grey triangle, left) and GABAergic interneurons (divided into nitric oxide ($N_{NO}$)- and neuropeptide Y ($N_{NPY}$)-expressing neurons; yellow oval shapes, right). These neurons are connected in a simple relationship, with pyramidal neurons ($N_{Pyr}$) exciting GABAergic neurons, which in turn inhibit the surrounding cells. Reduction of excited neural activity is illustrated as a flux towards $\emptyset$, likewise, mass removal from other states is illustrated with a flux towards $\emptyset$. Activation of a neuron causes an influx of $Ca^{2+}$ ions, which is initiated by the electrical stimulation described by $u_1$, $u_2$, and $u_3$ for the respective neurons. In pyramidal neurons, $Ca^{2+}$ activates phospholipases, which convert phospholipids into arachidonic acid (AA). AA is further metabolized into prostaglandin E2 ($PGE_2$). In NO-expressing interneurons, $Ca^{2+}$ activates nitric oxide synthase, which triggers the production and release of nitric oxide (NO). In other subtypes of GABAergic interneurons, vesicle-bound vasoactive peptides such as neuropeptide Y (NPY) are expressed. The release of these peptides is facilitated by $Ca^{2+}$. Vascular smooth muscle (VSM) cells (brown rectangles) enwrap arterioles (left side of the vascular tree) and regulates the arteriolar diameter. $PGE_2$ promotes arteriolar dilation by activation of the prostaglandin EP$_4$ receptor located on the VSM. NO diffuses freely over cellular membranes and acts to increase the

production of cyclic guanosine monophosphate, which in turn promotes arteriolar dilation. Lastly, NPY activates the G-protein coupled NPY Y1 receptor (NPY1R) expressed on VSM cells, promoting arteriolar constriction. These three effects on the VSM control the arteriolar diameter and thereby also the volume ($V_A$) and flow of blood ($f_{A,C}$) through the arterioles. These volume and flow changes are propagated through the capillaries ($V_C$, $f_{C,V}$) and venules ($V_V$, $f_{out}$). B: Three-compartment vascular model of blood volumes ($V_A$, $V_C$, $V_V$), pressures, and flows ($f_{in}$, $f_{A,C}$, $f_{C,V}$, $f_{out}$) corresponding to an analog electrical circuit, as described in [34]. The blood pressure drop corresponds to a voltage drop, the blood flow to an electric current, and the blood volume to electric charge stored in the capacitors. The vessel compliance ($C_1$, $C_2$, $C_3$) plays the role of capacitance, and the vessel resistance ($R_1$, $R_2$, $R_3$) is analogous to electric resistance. The blood pressure difference ($\Delta p_r$) maintained by the circulatory system corresponds to the electromotive force. C: Oxygen transport model, as described in [43]. The diagram depicts amount of oxygen ($n_iO_2$, i = {A, C, V, t}), oxygen saturation ($S_iO_2$, i = {A, C, V}), oxygenated hemoglobin ($HbO_i$, i = {A, C, V}) and deoxygenated hemoglobin ($HbR_i$, i = {A, C, V}), for each respective compartment (A = arterial, C = capillary, V = venous, t = tissue). Oxygen in tissue can be metabolized, indicated by the cerebral metabolism of $O_2$ ($CMRO_2$) arrow leaving the state. These different models in unison affect the blood oxygenation and blood volume in each respective compartment, which in turn determines the specific compartments contribution ($S_i$, i = {A, C, V, E}) to the BOLD-fMRI signal (grey boxes), where $S_E$ is the oxygen saturation of the extravascular tissue.

ms duration), and as in Barrett *et al.* [34], we opted to implement this paradigm as one constant stimulation block (i.e. a square wave, combining the period between each air puff, 105 ms, and the air puff itself, 20 ms, with 8 puffs a second equalling 1000ms) to reduce the complexity of the stimulation input to the model. The resulting stimulation has a constant amplitude with the following durations: 125 ms, 10 s, and 30 s. In the model, we calculate the diameter change as the square root of the volume change for each respective compartment. The model parameters were fitted to all experimental time series simultaneously (Fig 3A–3F, colored symbols), achieving a quantitative acceptable agreement to experimental data (Fig 3A–3F, solid, colored lines), evaluated using a $\chi^2$-test ($J_{lsq}(\hat{\theta}_{H1})$ = 292.59, cut-off: $\chi^2$ (288 data points) = 328.58 for $\alpha$ = 0.05). The degrees of freedom for the confidence interval were $df_{H1}$ = 37, which is equal to the number of estimated parameters. The reader is referred to S1 Appendix for the posterior probability profiles (S1 Fig) and the parameter boundaries applied during the estimation (S1 Table). The estimated model uncertainty is depicted in the form of shaded red and blue areas in Fig 3.

The simulations follow the qualitative behavior of the experimental data for each graph except for Fig 3E, which is the venular response to a 10 s stimulation, where the model simulations show a slower rise and decline than the experimental data (Fig 3E, compare blue line with blue symbols). A more accurate fit can be obtained by allowing the viscoelasticity and stiffness coefficients of the capillary and venous compartments to change between the two short (125 ms & 10 s) and the long (30 s) stimulation (S2 Fig, $J_{lsq}(\hat{\theta}_{H2})$ = 199.69 with the parameter profiles depicted in S3 Fig). This improved agreement indicates that a non-linear vascular model is needed to fully capture the dynamics seen in this data set.

**2.1.2 The observed dynamic behavior can be explained by two distinct dilation processes.** The arteriolar responses transition from a one-peak (Fig 3A) to a two-peak response (Fig 3C), as the stimulation length increases. In other words, for the short stimulation, the response rises to a peak after 2–3 seconds, followed by a decline to baseline. For the two longer (10 s and 30 s) stimulations, a quick initial dilation is followed by a small cessation to a plateau value around 10 s (Fig 3B and 3C). At 10 s, the stimulation ends in one case, with a minor dilation before returning to baseline diameter (Fig 3B). However, for the 30 s stimulation, a new dilation phase occurs, consistently dilating until the end of the stimulation (Fig 3C, t = 10–30 s). Finally, after stimulus cessation for the longest stimulation, but not for the shorter stimulations, a post-peak undershoot in the arteriolar diameter is observed before returning to baseline (Fig 3C, t ~ 50 s).

These complex dynamic behaviors between different stimulation lengths are captured by the model (Fig 3A–3C). It is therefore relevant to examine by which mechanisms the model produces these behaviors. Model simulations are shown for all three stimulation lengths: 0.125 s (Fig 4A & 4E), 10 s (Fig 4B & 4F), and 30 s (Fig 4C & 4G). Furthermore, Fig 4A–4C correspond to the rapid neuronal responses (Fig 4A–4C), and Fig 4E–4G correspond to the impact

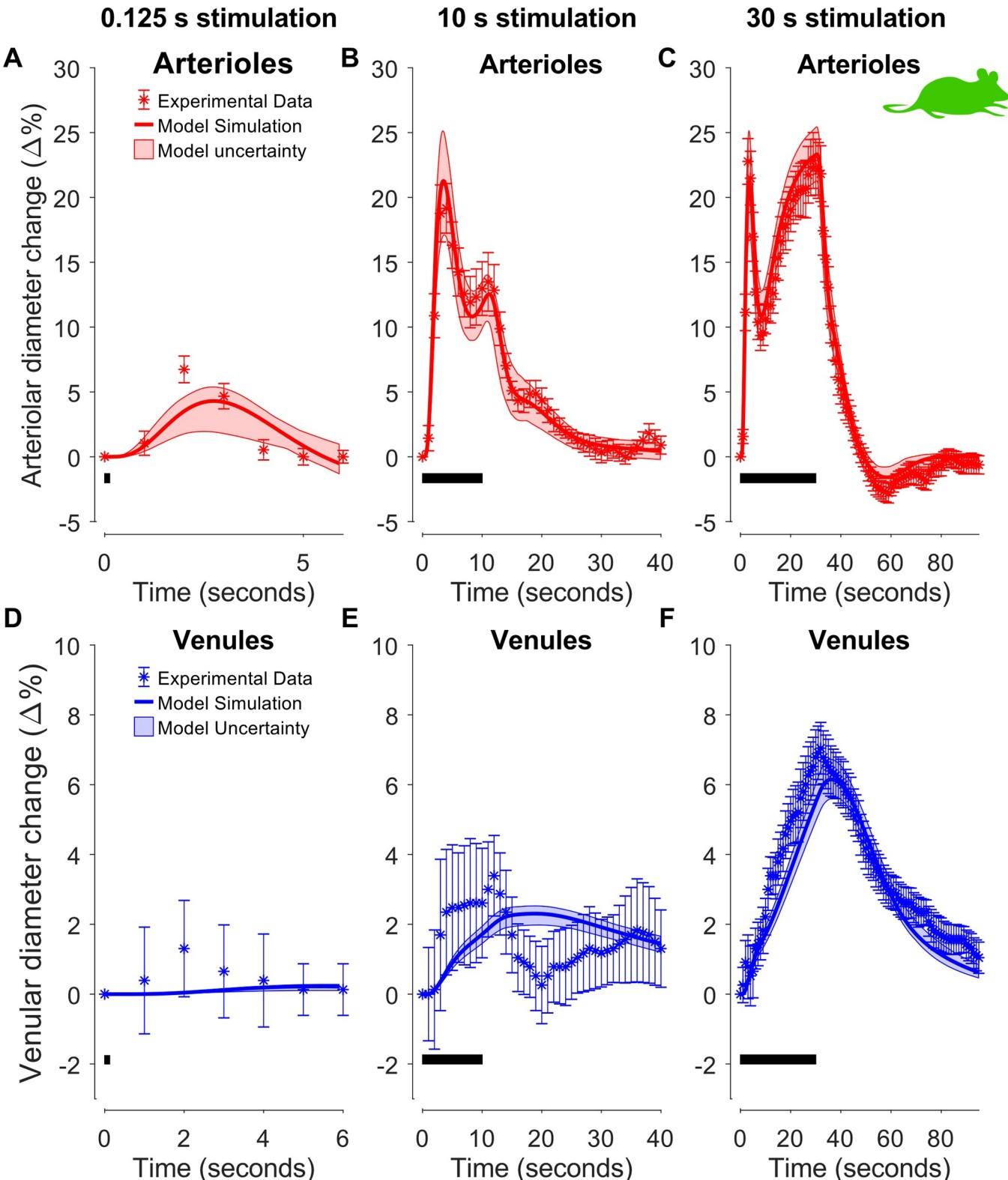

**Fig 3. Model estimation to experimental data in Section 2.1.1.** Data and simulations for arteriolar (A–C) and venular (D–F) diameter changes in awake mice for three different sensory stimulation lengths: 125 ms (A & D), 10 s (B & E), and 30 s (C & F). Experimental data are replotted versions of data presented in Fig 2C of the original manuscript by Drew et al. [46]. The stimulation lengths are denoted with the black bar in the bottom left portion of each graph. For each graph: experimental data (colored symbols); the uncertainty of the experimental data is presented as the standard error of the mean (SEM) (colored error bars);

the best model simulation is seen as a colored solid line; the model uncertainty as a colored semi-transparent overlay. The x-axis represents time in seconds, and the y-axis is the normalized vessel diameter change (Δ%).

of the vasoactive substances: NO (nitric oxide), $PGE_2$ (prostaglandin E2), NPY (neuropeptide Y). In addition, the simulations are shown for all three neuronal types and corresponding vasoactive substances: pyramidal neurons and $PGE_2$ (orange), NO interneuron and NO (light green), and NPY interneurons and NPY (green-gray). Finally, the resulting behavior (Fig 4E–4G, black dashed line) is produced by the summation of the three impacts, where NPY is vaso-constrictive (negative), and $PGE_2$ and NO are vasodilative (positive). To assist navigation between these different simulations, Fig 4D gives a simplified view of the model and the mechanisms which underlie the different dilation behaviors observed in Fig 3A–3C.

With these interpretations in place, we can understand how the model produces the complex dynamic behaviors observed in data. It is easiest to follow the chain of events by starting at the last step before the observed behavior. The response to a short stimulation is produced by the fast release of NO, which produces the initial peak in the arteriolar response (Fig 4F–4G, light green line) and which thereafter simply declines. Note that the vascular effect is the relative difference of released NO compared to the initial condition and a negative value therefore should be interpreted as lower levels compared to the basal state and not as negative (see Eq 8). The other two signaling pathways are slower and therefore negligible. Also, for the longer stimulations, the NO behavior is important. For these stimulations, the NO response culminates quickly, whereafter the impact of NO falls which causes a decrease after the first peak. However, for longer stimulations, this decrease is replaced by a new increase, which is caused by the rise of the slower $PGE_2$ signaling arm from the pyramidal neuron (Fig 4F–4G, orange line). In other words, the $PGE_2$ arm is responsible for the second dilation phase observed for the 30 s stimulation (Fig 4G, orange line). Finally, the NPY signaling arm is only important to explain the post-stimulus undershoot observed in the 30 s stimulus paradigm as it is so slow that it requires such long stimulation to become larger than the two other arms (Fig 4G, the green-gray line is above the light green and orange for t ~ 50 s). In summary, the initial peak is explained by the fast NO arm, the second peak is explained by the slower $PGE_2$ arm, and the post-peak undershoot is explained by the even slower NPY arm.

These different dynamics in the three arms occur at the downstream secretion level but must be initiated by corresponding changes at the rapid neuronal level (Fig 4A–4C). For the shortest stimulation (Fig 4A), the NO interneurons are the fastest, and pyramidal neurons and NPY interneurons are equally fast. For the two longer stimulations (Fig 4B and 4C), the electrophysiology reaches a steady-state within 1 second, whereafter no changes occur until the stimulation ends. Since the secretion of NPY is much slower than $PGE_2$, and because both these dynamics occur over many seconds, the differences in dynamics at the downstream secretion level are almost exclusively explained by the difference in intracellular signaling. The only difference concerns the NO dynamics, which has a peak and decline at the secretion level caused by a corresponding rise and fall at the electrophysiology level.

**2.1.3 The model can still describe and correctly predict arteriolar responses to optogenetic and sensory stimulations.**   In our previous computational work (see [42]), we developed a model capable of estimating and predicting the experimental data found in Uhlirova *et al.* [47]. To verify that this extended model still maintains a good agreement and predictive power with that experimental data, we repeated the model training and model predictions in [42], see S4 Fig. The Uhlirova study reports arteriolar diameter changes in mice evoked by different stimuli. These different stimuli include for instance optogenetic and sensory stimulation, where optogenetics can stimulate either pyramidal neurons or inhibitory interneurons

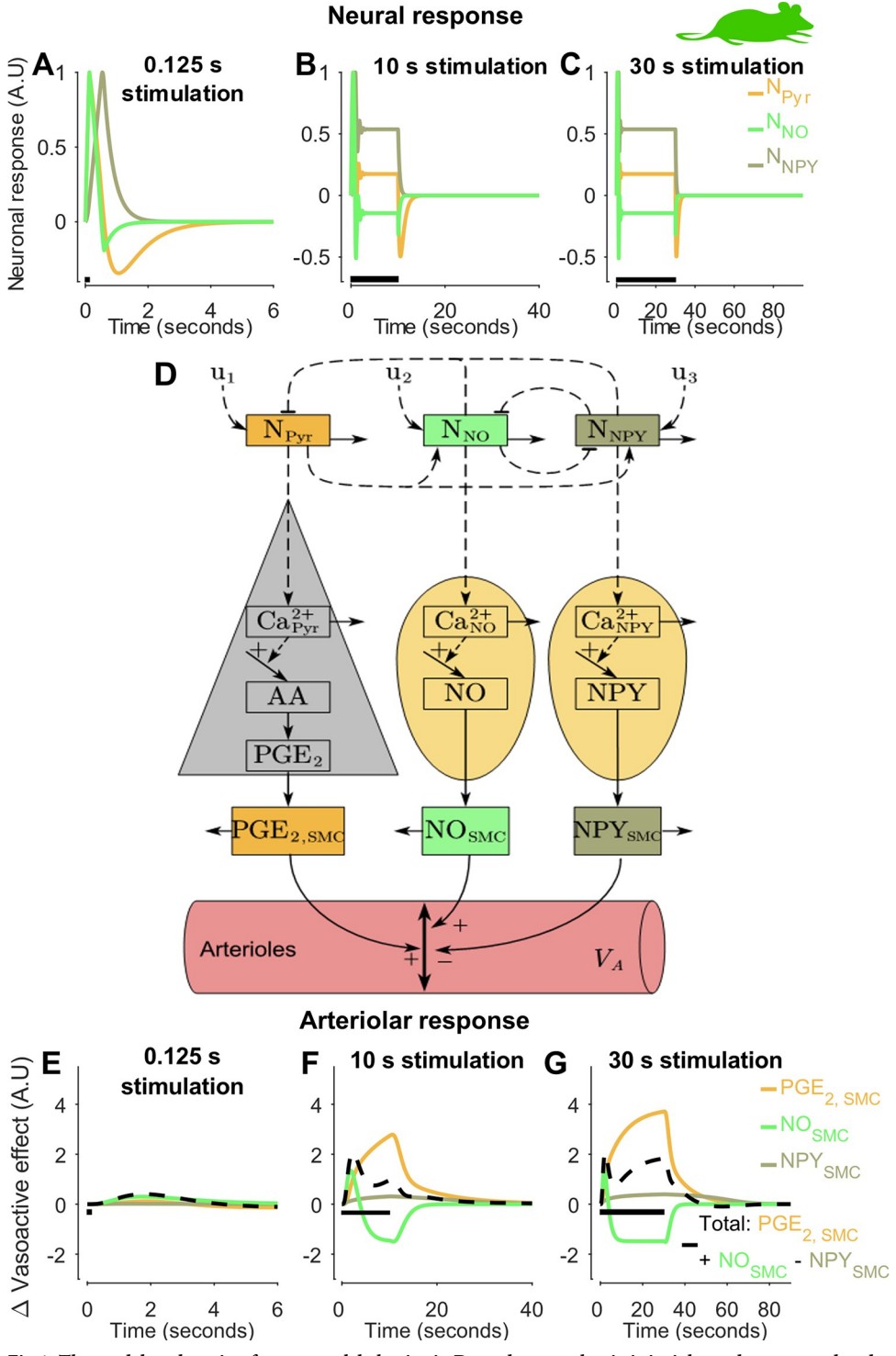

**Fig 4. The model explanation for two-peak behavior in Drew data: mechanistic insights to be preserved and translated (Section 2.1.2) [46].** Model simulations for the three stimulus durations: 125 ms (A & E), 10 s (B & F), and 30 s (C & G) are shown. For each stimulus: the dynamic of the neuronal states, pyramidal neurons ($N_{Pyr}$), GABAergic nitric oxide interneurons ($N_{NO}$), GABAergic neuropeptide Y interneurons ($N_{NPY}$), (A–C), and the arteriolar response (E–G). For each graph: model simulation (colored lines); stimulus length (black bar). The x-axis represents time in seconds, and the y-axis is the change in neuronal states (A.U) for A–C, and the vasoactive effect on the arteriolar compartment (A.U) for E–G. In D, a simplified overview of the model is given. Here, $u_1$, $u_2$, and $u_3$ are the stimulus input to the model and are applied for 125 ms, 10 s, and 30 s for each respective experimental setting. The relative

timing and function of the three arms ($PGE_{2, SMC}$, $NO_{SMC}$, and $NPY_{SMC}$) depicted in this figure is the first mechanistic insight to be translated to the analysis of the other datasets from the other species (Figs 1E and 5–7). Note that the vasoactive effect presented in E-G is expressed from the relative change from the baseline, see Eq 8, and therefore negative values should be interpreted as lower concentration compared to a non-stimulated state.

separately. They do this cell-specific stimulation with and without pharmacological perturbations, using the NPY receptor inhibitor BIBP, and the glutamatergic signaling inhibitors CNQX and AP5. These pharmacological perturbations cut off the crosstalk between NPY and the vasculature and the crosstalk between pyramidal neurons and interneurons, respectively. Furthermore, they also study the animals during awake and anesthesia conditions. In [42], our model can explain all these data with the same parameters, and can also predict independent data not used for training. Our updated model has the same capability (see S1 Appendix section 1.3), and these results *e.g.*, support the notion that NPY acts vasoconstrictive, and causes the post-peak undershoot.

**2.1.4 Hemoglobin and BOLD measures in awake mice.**   Next, we analyzed experimental data from the study of Desjardins *et al.* [48]. In brief, the study reports hemodynamic measures of blood oxygenation (oxygenated, deoxygenated, and total hemoglobin) from three different stimulations: 1) optogenetic stimulation of inhibitory interneurons; 2) optogenetic stimulation of pyramidal neurons, and 3) sensory whisker stimulation using air puffs. Each of these three paradigms had a short (optogenetic: 100 ms light pulse; sensory: 2 s of air puffs at 3–5 Hz) and long (optogenetic: 20 s of 100 ms light pulses at 1 Hz; sensory: 20 s of air puffs at 3–5 Hz) duration. They also reported BOLD data for the long optogenetic stimulation of pyramidal neurons. We implemented these stimulus paradigms in the model and assumed a 100 ms air puff delivered at 3 Hz for the sensory stimulus. To capture the variability in frequency, which gives rise to an observable difference in stimulation strength between the two sensory experiments (Fig 5C and 5F), we allowed the input parameters ($k_{u,i}$) to change between the short and long duration of the sensory stimulation but could keep the same values for both optogenetic paradigms (see Discussion 3.3). Furthermore, in the model, we assumed a baseline blood concentration of 100 μM hemoglobin, in line with previous models [35,37,50].

With these data and model settings, the model was trained to experimental data of hemoglobin dynamics for all three stimulation paradigms simultaneously (Fig 5A–5F, dark-red/light-blue/green symbols), and the BOLD data was left out to be used for model validation (Fig 5J, magenta symbols). The model achieved a quantitative acceptable agreement to experimental data (Fig 5A–5F, red/blue/green lines) after parameter estimation ($J_{lsq}(\hat{\theta}_{H3})$ = 323.36, cut-off: $\chi^2$ (354 data points) = 397.87. The degrees of freedom for the confidence interval was $df_{H4}$ = 43, which is equal to the number of estimated parameters (Fig 5A–5F, shaded areas). The reader is referred to S1 Appendix for the posterior probability profiles (S5 Fig) and the parameter boundaries applied during the estimation (S3 Table)

To validate the model and avoid overfitting of the parameters, we used the model to generate predictions of a BOLD response for a 20 s optogenetic stimulation of pyramidal neurons. These predictions are depicted as the magenta-shaded area in Fig 5J. The corresponding experimental data (Fig 5J, magenta symbols) lie within the model prediction bounds. The parameter that best describes training data also passes a $\chi^2$ test for the BOLD validation data ($J_{lsq}(\hat{\theta}_{H3})$ = 29.23, cut-off: $\chi^2$ (23 data points) = 35.17).

## Preservation of qualitative features: order and function of the three signaling arms

Finally, we preserved the mechanistic insights obtained previously (Fig 5G–5I). More specifically, for the sensory stimulation (Fig 5I), we required that the order and relative function of

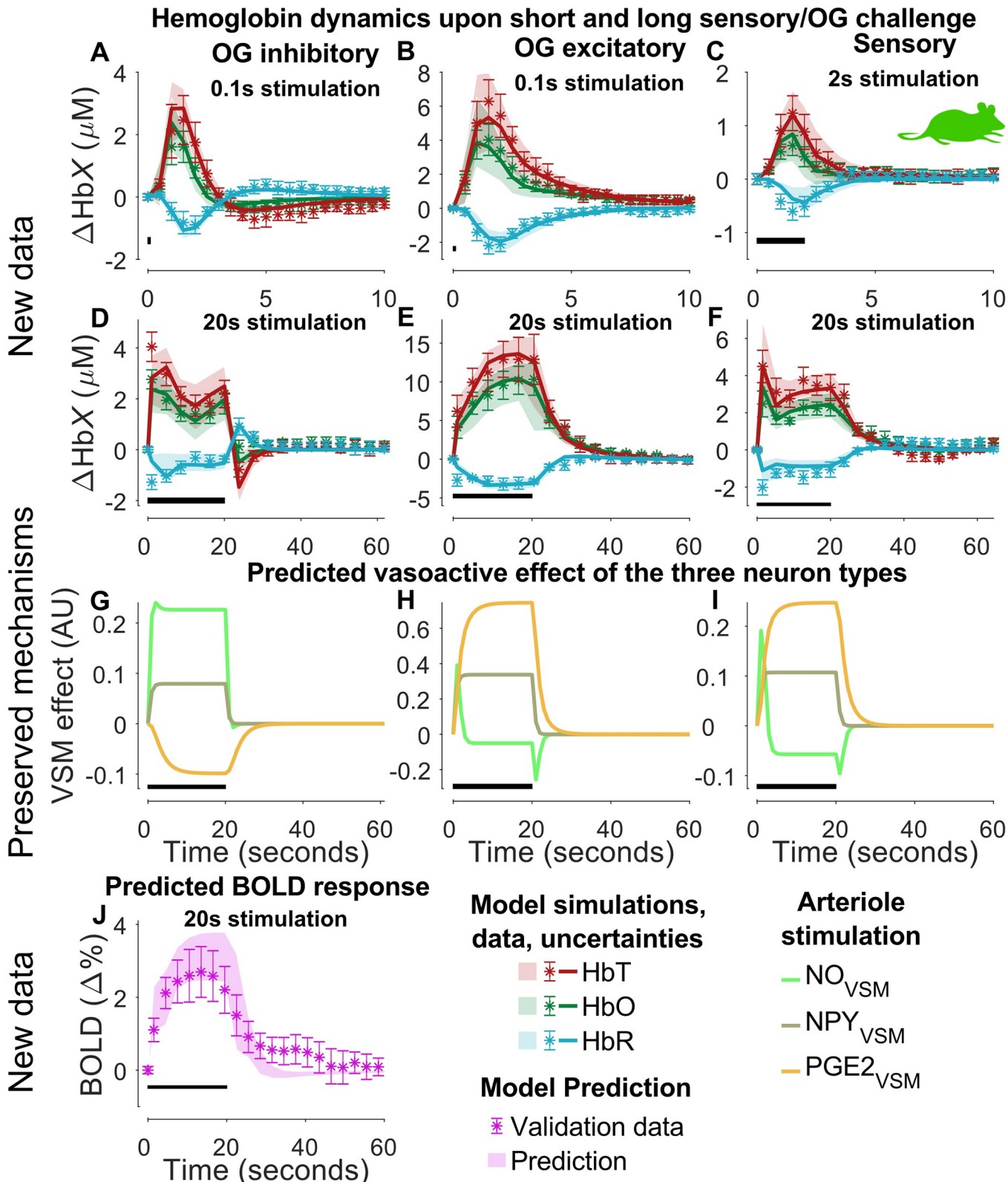

**Fig 5. Model estimation to experimental data of hemoglobin changes in mice for three different stimulation types (Section 2.1.4).** The experiment features optogenetic (OG) activation of inhibitory (A & D) and excitatory (B & E) neurons, and sensory stimulation (C & F). For each stimuli type, a short stimulus (OG: 100 ms of light (A-B); Sensory: 2 s I) and a long stimulus (OG: 20 s (D-E); Sensory: 20 s (F)) was used. This is denoted with the black bar in the

bottom left portion of each graph. The shown experimental data are replotted versions of S5 Fig of the study by Desjardins et al. [48]. For A-F: experimental data consisting of oxygenated hemoglobin (HbO; dark-red symbols), deoxygenated hemoglobin (HbR; light-blue symbols), and total hemoglobin (HbT; green symbols); the uncertainty of the experimental data is presented as the standard error of the mean (SEM) (colored error bars); the best model simulation is seen as colored solid lines corresponding to respective measurement variable; the model uncertainty as colored semi-transparent overlays; the x-axis represents time in seconds, and the y-axis is the change in hemoglobin concentration (μM). G-I: preserved mechanisms i.e., correct relative timing, of the vasoactive effect on the VSM from the different vasoactive substances, nitric oxide (NO), neuropeptide Y (NPY), and prostaglandin E2 ($PGE_2$). J: model predictions (shaded area) and experimental data (mean ± SEM, symbols) of a BOLD response to an identical stimulus is shown in E. The experimental data were extracted from Desjardins et al. 2019 [48].

the three signaling arms should be the same as those identified in the previous rodent data (Fig 4): NO (light-green) displays the fastest initial response followed by a rapid decline and $PGE_2$ (orange) displays a slower but more prolonged response covering the main part of the NVC control. The NPY (green-gray) response is slow, but since there is no post-peak undershoot seen in the vascular response, the NPY dynamics do not have to be as slow as in Figs 3 and 4. Note that the direct timing requirement from Figs 3 and 4 only holds for the sensory stimulation, since sensory stimulation was applied for data in Figs 3 and 4. Nevertheless, we required that the other two optogenetic responses still preserve some basic insights e.g., that NO and $PGE_2$ responses are vasodilating via an initial increase and not a decrease. Note also that the $PGE_2$ response to optogenetic-stimulation of interneurons (Fig 5G, orange) is negative, which is expected since the pyramidal cells have not received any direct stimulation, but only negative feedback from the interneurons.

**2.1.5 Summary of all mice data.** In summary, all results up until now taken together show that our model structure provides an acceptable explanation for a wide variety of NVC aspects observable in mice. Figs 3 and 4 unravel the role of three regulatory arms: NO interneurons are responsible for the rapid rise, $PGE_2$ from pyramidal neurons are responsible for the second peak that occurs for long stimulations, and NPY interneurons for the post-peak undershoot. In these data, vessel diameter is used as a proxy for the NVC control. In Fig 5, we confirm that our model can explain the NVC control expressed in hemoglobin measures: total hemoglobin serves as a proxy for blood volume, oxygenated- and deoxygenated hemoglobin examines the balance between metabolism and blood flow, and the BOLD signal provides a link to the most common non-invasive measure in primates and humans. For a comparison of the posterior probability profiles for Figs 3 and 5 –see S1 Appendix section 1.7.1. Finally, just like our previous model [42], our new model can explain the highly informative optogenetic-data from Uhlirova *et al.* (2016) [47].

## 2.2 Neuronal and BOLD measures in anesthetized macaques

**2.2.1 The model agrees with BOLD and LFP data.** We analyzed experimental data from the study of Shmuel *et al.* [10], which presents unique data for higher primates *i.e.*, macaques. We extracted electrophysiological and BOLD responses induced by a visual task in anesthetized macaques. The task was presented in two ways, inducing both positive and negative responses in the same area (see section 4.4.3 for details). The electrophysiological response was related to the local field potential (LFP), which is to a large extent influenced by pyramidal neuron firing (see review in [51]). The model was trained to experimental data for both positive (Fig 6A and 6B, BOLD, magenta and LFP, pink symbols) and negative (Fig 6A and 6B, BOLD, purple and LFP, dark-purple symbols) responses simultaneously. The different experimental stimulus was applied as a square pulse to the model (see Eq 2). For the negative experimental setup, sign parameters were added to interpret if the stimulus should act positively or negatively upon the neurons (see Eq 3), resulting in a negative square pulse. The model achieved a quantitative acceptable agreement to experimental data (Fig 6A and 6B, red and

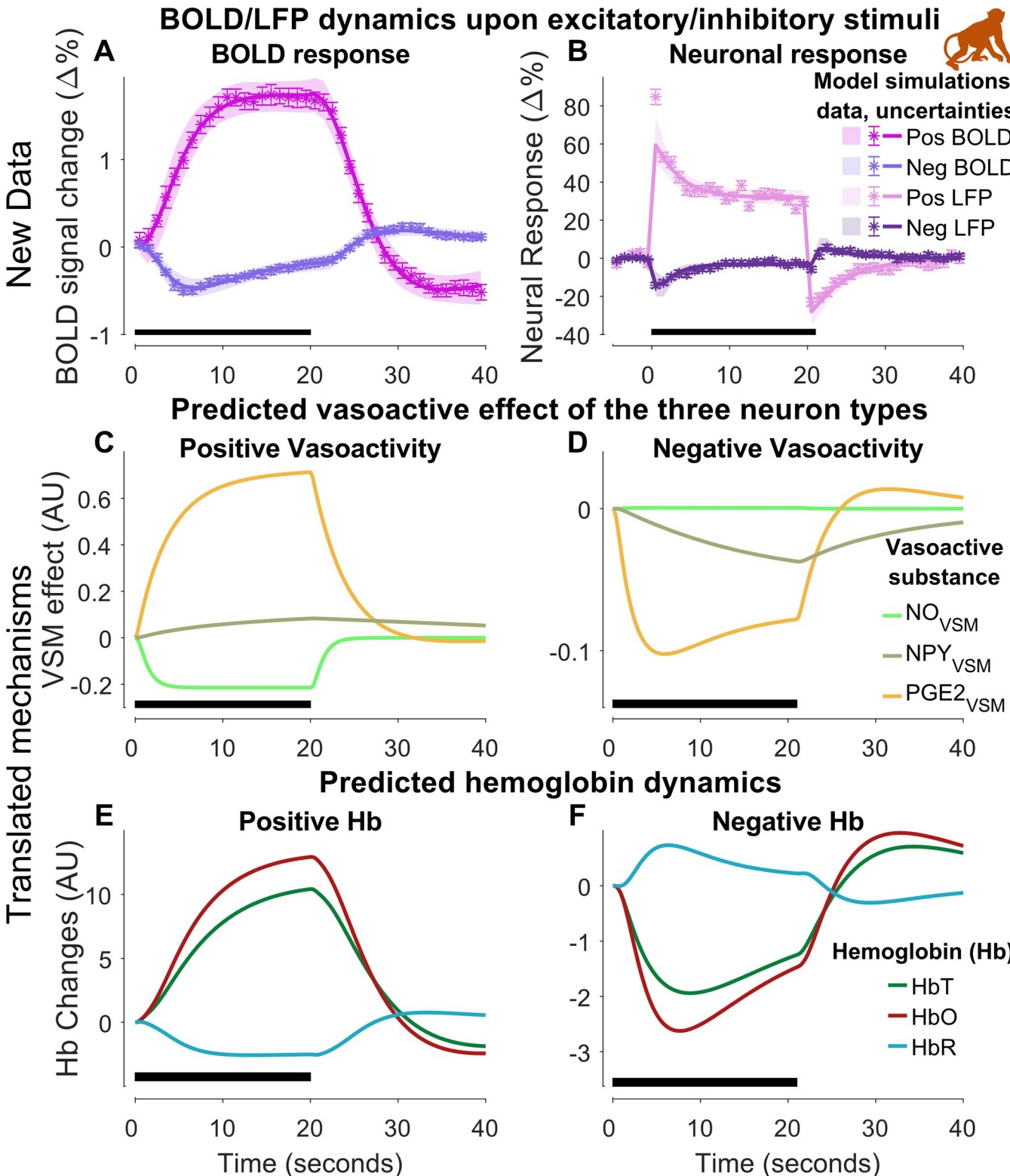

**Fig 6. Model estimation to experimental data from macaque primates (Section 2.2).** We see blood oxygen level dependent (BOLD) (A) and electrophysiological (B) responses in macaques for two variants of a visual task. The visual task was 20 s long, for the positive task, and 21 s, for the negative task, visual stimulation of V1 in the visual cortex (marked with the black bar in the bottom left portion of each graph). In A & B: experimental data (colored "∗"-symbols); the uncertainty of the experimental data is presented as the standard error of the mean (SEM) (colored error bars); the best model simulation is seen as colored solid lines corresponding to respective measurement observables. The model uncertainty is seen as the semi-transparent colored areas. The x-

axis represents time in seconds, and the y-axis is BOLD signal change (Δ%) for A and local field potential (LFP) expressed as percent change from baseline (Δ%) for B. C-D: translated dynamics and mechanisms regarding the effect on the VSM from the different vasoactive substances: nitric oxide (NO), neuropeptide Y (NPY), and prostaglandin E2 (PGE$_2$) given as arbitrary units (A. U.). E-F: translated dynamics from mice regarding the hemoglobin changes given as arbitrary units (A. U.).

blue solid lines) after parameter estimation ($J_{lsq}(\hat{\theta}_{H4})$) = 176.13, cut-off: $\chi^2$ (160 data points) = 190.52). The degrees of freedom for the confidence interval were $df_{H4}$ = 52, which is equal to the number of estimated parameters (Fig 6A and 6B, shaded areas). The reader is referred to S1 Appendix for the posterior probability profiles (S6 Fig) and the parameter boundaries applied during the estimation (S4 Table).

**2.2.2 The model translates the mechanistic insights and qualitative behaviors obtained from mice data.** Finally, we translated the mechanistic insights obtained from studies in mice (Figs 3–5) to this analysis on primates (Fig 6C–6F). More specifically, for the sensory stimulation (Fig 6C), we know aspects of the order and relative function of the three signaling arms (Fig 4): NO (light-green) displays only a fast initial response followed by a rapid decline; PGE$_2$ (orange) displays a slower and more prolonged response covering the main part of the NVC control; NPY (green-gray) has the slowest response, and is only prominent in the creation of the post-peak undershoot. Here, there is no visible peak on NO, and the response almost directly goes to the rapid decline. This behavior is needed for the model to be able to generate the rapid response in LFP, which is predominantly caused by the pyramidal cells, which are inhibited by the NO-interneurons. In other words, if the NO-interneurons would have had a substantial rapid increase, the LFP simulation would not be fast or high enough. Note that there is no known information about the relationship between the three arms for the negative BOLD signal (Fig 6D). Apart from these mechanisms, we also translated mechanistic knowledge from Fig 5, concerning the relative dynamics of total hemoglobin (HbT), oxygenated hemoglobin (HbO), and deoxygenated hemoglobin (HbR) (Fig 6E and 6F). Just as in the mice data, for the positive BOLD response, the model is showing an increase in HbT (green), an even greater increase in HbO (dark red), and a decrease in HbR (light blue), all with the same dynamics as in the BOLD response. For a further comparison of the posterior probability profiles for Figs 3–7 –see S1 Appendix section 1.7.2.

## 2.3 MR-based hemodynamic measures in awake human

**2.3.1 The model agrees with both estimation and validation data for CBV, CBF, and BOLD data, from both arterioles and venules.** In the last step, we move to humans and looked at the study conducted by Huber *et al.* [49]. We extracted the experimental measurements of CBV, CBF, and BOLD responses in humans, evoked by multiple visual stimuli. The study employed a set of three different flickering checkerboard patterns as the visual stimuli (see section 4.4.5. for more details). The model was trained on experimental data consisting of both positive and negative CBV (Fig 7A) and BOLD responses (Fig 7D), as well as experimental data for the total CBV changes for the excitatory (Fig 7B, black) and inhibitory (Fig 7C, black) tasks. Parameter estimation was employed to achieve an acceptable fit to the data ($J_{lsq}(\hat{\theta}_{H6})$) = 54.64, cut-off: $\chi^2$ (122 data points) = 148.78). The reader is referred to S1 Appendix for the posterior probability profiles (S7 Fig) and the parameter boundaries applied during the estimation (S5 Table). The simulation uncertainty can be seen as the colored shaded areas in Fig 7A–7E. As seen, the model simulations have a good agreement with the experimental data (Fig 7A and 7D, shaded areas, and Fig 7B and 7C, black shaded areas), as the model uncertainty area overlaps well with the experimental data points. To test the model's predictive qualities, we used the model to generate predictions of CBF for both the positive and negative

 

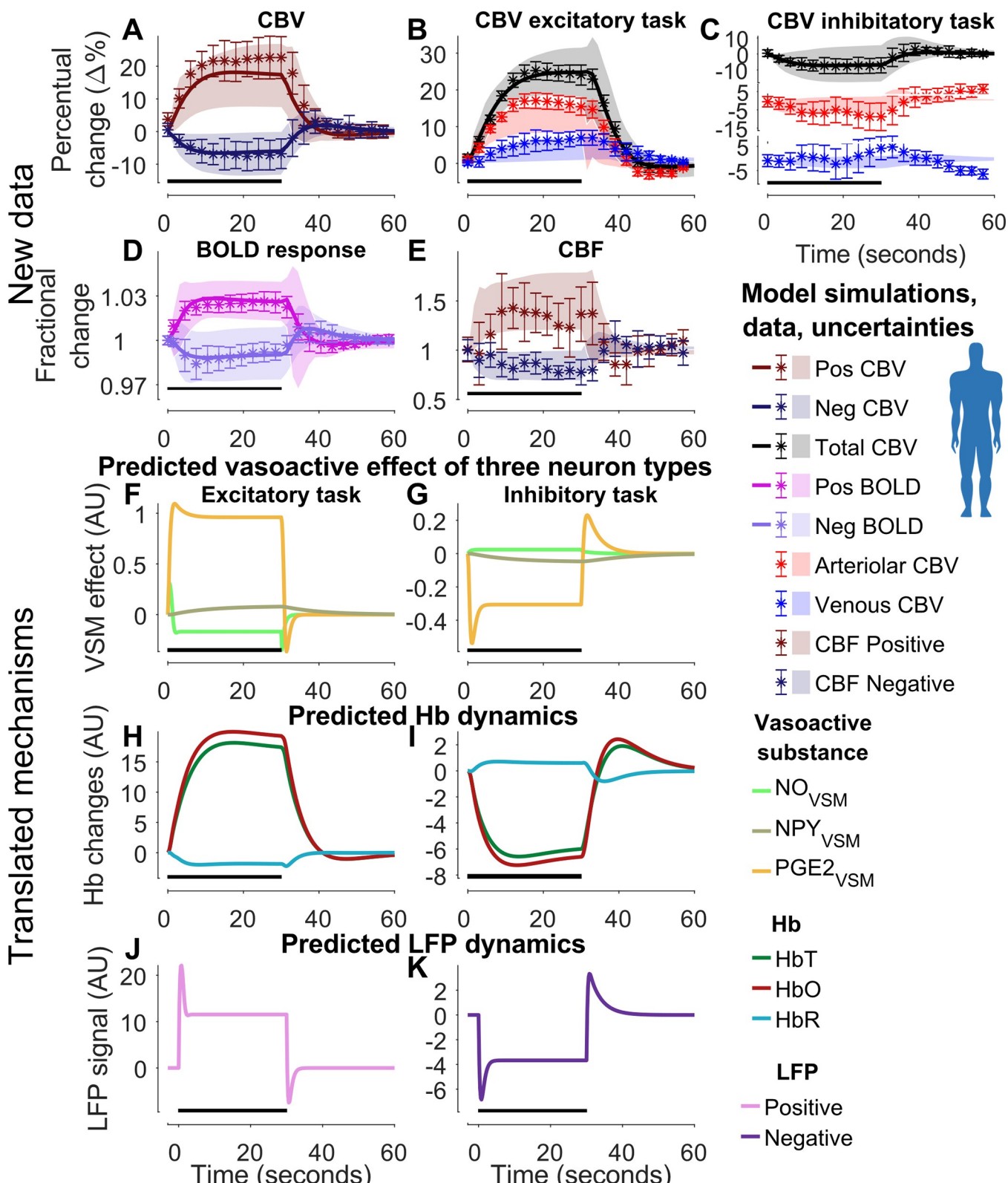

**Fig 7. Model estimation and prediction of MR-based experimental data in humans for different visual stimulation tasks (Section 2.3).** The graph contains data for three different tasks: cerebral blood volume (CBV) (A), blood oxygen level dependent (BOLD) signal (D), and cerebral blood flow (CBF) (E) changes for small flickering checkerboard task, and compartment-specific CBV changes for an excitatory (B) and inhibitory (C) flickering tasks. For each graph, the

stimulus duration (30 s) is depicted as the black bar in the bottom left portion of each graph. In A-E: experimental data shown with the measurement uncertainty as the standard error of the mean (SEM) (colored error bars); the best model simulations are seen as colored solid lines corresponding to respective measurement observables; the model uncertainty is depicted as colored semi-transparent overlays; the x-axis represents time in seconds, and the y-axis is the change in fractional (D, E) or percentual change (A–C) of the measurement observables. The model prediction uncertainty of CBF (E) and compartment-specific CBV changes (B, C) is depicted as semi-transparent overlays. The experimental time series are taken from Huber et al. [49]. Translated mechanistic insights are shown for the three arms of vascular control (F, G), hemoglobin dynamics (H, I), and local field potential (LFP) (J, K), for the positive (F, H, J), and negative (G, I, K) BOLD response. F-K are given as arbitrary units (A. U.).

responses (Fig 7E), as well as predictions of arterial CBV and venous CBV for both the excitatory (Fig 7B, red and blue) and inhibitory flickering checkerboard tasks (Fig 7C, red and blue). As can be seen, there is a good agreement between the model predictions (Fig 7B, 7C and 7E, areas without a line) and the experimental data (error bars).

**2.3.2 The model has translated all insights obtained from both rodent and primate data.**   We translated all mechanistic insights obtained so far i.e., from both the mice (Figs 3–5) and the primate (Fig 6) data. For the positive BOLD response (Fig 7D), we know and preserve the order of the three arms: the NO-response (light green) is fastest and followed by a rapid decline, the $PGE_2$ response (orange) is slower and covers the main response, and the NPY-response (grey-green) is slowest and is only dominant during the post-peak undershoot (Fig 7F). For the hemoglobin dynamics (Fig 7H) in the positive BOLD response, we know that the total hemoglobin (HbT, green) increases due to an increased blood flow, which increases oxygenated hemoglobin (HbO, dark red) and decreases deoxygenated hemoglobin (HbR, light-blue) (Fig 5F). Finally, from the primate data (Fig 6), we know that the LFP response (Fig 7J) is faster than the BOLD response, has an overshoot that goes down to an intermediate plateau throughout the stimulation, and has a corresponding undershoot below baseline post-stimulation. We show corresponding predictions for these three properties (the three arms of vascular control, Fig 7G; hemoglobin dynamics, Fig 7I; and LFP, Fig 7K), but it is only for LFP that a known behavior is translated. For a further comparison of the posterior probability profiles for Figs 3–7 –see S1 Appendix section 1.7.2.

## 3. Discussion

Currently, no existing model of the NVC can simultaneously describe the large variety of available NVC data (Fig 1B and 1C): i) BOLD, ii) CBV, iii) CBF, iv) Hemoglobin measures, v) optogenetic and sensory stimulations, vi) pharmacological perturbations including the effect of anesthetics (see S1 Appendix section 1.3), and vii) phenomenological description of LFP-related signals. Herein, we provide a first such model (Fig 2). The model can quantitatively explain experimental data used for model training (Figs 3, 5, 6, 7 and S2A–S2F) and correctly predict experimental data not used for training (Figs 5J; 7B, 7C, and 7E), from rodents (Figs 3, 5 and S2), primates (Fig 6), and humans (Fig 7). This validated model provides a first interactive and inter-connected explanation for how the complex dynamics and interdependencies in these data can be understood. We illustrate this potential by using the model to obtain mechanistic insights from each dataset considered individually, and these insights are then translated to the analysis of the other species and datasets. The first insight concerned the timing, relative amplitude, and function of the three signaling arms, which were obtained in the analysis of the double peak response in arteriolar diameter in response to prolonged stimuli (Fig 4G, black broken line); the first peak is produced by the rapid NO dilating arm, which has a rapid subsequent decline (Fig 4E–4G, light-green line), the second peak is produced by the slower $PGE_2$ arm (Fig 4E–4G, orange line), and the post-peak undershoot is explained by the even slower NPY arm (Fig 4E–4G, gray-green line). The second insight concerned the dynamics of

hemoglobin changes (Fig 5): following the increase of blood flow, oxygenated hemoglobin increases, as the supply of oxygen far outweighs the consumption. Coupled with this, the amount of deoxygenated hemoglobin decreases, as the concentration of oxygenated hemoglobin increases, leading to a total increase of hemoglobin. The third insight concerned the relationship between LFP and BOLD, which comes from the analysis of the primate data (Fig 6): LFP shows a more rapid increase than BOLD and is followed by an overshoot down to a lower plateau throughout the duration of the stimulation, and a corresponding post-peak undershoot. Finally, the model combines all these mechanistic insights (Fig 7F–7K), with those specific to the human data, which unravels the relationship between BOLD, CBF, and CBV in arterioles and venules (Fig 7). Our new model serves as a foundation for a new more integrative approach to the analysis of neuroimaging data, which opens the door to many new applications in both basic science and the clinic.

### 3.1. Relation to previous models of the NVC

The NVC has previously been mathematically modeled using different approaches. The earliest models, which still are used in the conventional analysis of BOLD-fMRI data today, are based on the general linear model (GLM), as implemented in software packages like statistical parametric mapping (SPM) [52,53]. The GLM model can identify activated brain areas but does not take biological understanding into account, and therefore the model cannot incorporate any additional data. Such incorporations require mechanistic models. Simple mechanistic models were presented by Friston *et al*. [27,54] and Buxton *et al*. [28,29]. These models are commonly referred to as the Balloon model. The Balloon model has become widely used, spawning many subsequent variants and improvements [36,55–58]. These models typically only describe venous volume changes (as this is the dominant contributor to the BOLD-fMRI signal). One significant shortcoming with these models is that they do not consider the arteriolar volume that is actively regulated in the NVC although primarily it is the arteriolar volume that changes upon short event-related stimuli (Fig 3). To remedy this, some researchers have developed analogous Windkessel models, which are electrical circuit equivalent models of pressures, volumes, and flows [34,35,59] that can describe all cerebrovascular volumes. In contrast, others have instead opted to exclusively describe arteriolar volume changes, which only are applicable to short stimuli [60,61]. Finally, other types of models are based on partial differential equations and detailed diffusion images [62–65]. A common shortcoming of all these mentioned approaches is that these models lack biochemical mechanisms underpinning the intracellular signaling, which translates neural activity to hemodynamic changes. Mathematical models describing these intracellular processes found in the NVC exist [38–41,66–68], but they are typically connected to the more simplified Balloon model, or lack realistic descriptions of the vascular dynamics. In conclusion, no model based on both biochemical mechanisms and vascular changes using a Windkessel model has previously been described (Fig 1C). In this work, we present a first such model.

### 3.2 Neuron module

In line with Sten et al. [42] we choose to include an excitatory neuron, pyramidal cells, and two types of interneuron, NO- and NPY-interneurons, in our module governing the vascular control, with PGE$_2$ and NO having a dilative effect and NPY having a constrictive effect. This is a simplification of reality since many more cell types and vasoactive substances are known [4–8]. From fitting our model to the vessel dilation data, expressing a double peak behavior [46] (Fig 4G), the model suggests a slow rising dilative effect from the pyramidal neurons and a fast depleting dilation response from the NO-interneurons. This is in line with previous

studies suggesting that the pyramidal cells are a major driving force in the dilatory response [47,69,70]. However, findings from recent studies are worth commenting on. An OG study [71] and a chemogenetic study [72] both showed that interneurons have a strong dilatory response on the arteries, in line with our model. Additionally, Echagarruga et al. found that solely chemogenetic activation of the pyramidal cells had little effect on the dilatory response, although, this was during a stimulation period of a few seconds. Lee et al. [73] showed that OG stimulation of NO- and somatostatin- (SST) interneurons showed similar, amplitude, and vascular response compared to sensory stimulation. Here, the stimulation was maintained for a longer duration of time and an argument for interneurons being the driving dilator and not having a depleting response. Although interesting findings, there is a difference between sensory and OG stimulation, and the study lack a stimulation control on the pyramidal cells that would be needed to rule out that the pyramidal cells have a little part in the dilation response.

### 3.3 Analysis of problematic data points

While the overall agreement between the model and data is statistically acceptable, as the model simulations pass $\chi^2$-tests, some individual data points cannot be explained by the model, and that warrants some additional comments. In section 2.1.1, we present the model estimation to experimental data collected from the original research in [46]. While the overall model agreement overlaps well with the experimental data, the model simulation does not agree well with data of venule dilation for the 10 s stimulus (Fig 3E, compare symbols with the shaded area). The experimental data exhibits a faster post-stimulus decay, while the venule dilation persists for a longer period in the model simulation. The identical behavior is seen in the original manuscript for the vascular model [34] (see Fig 2F in [34] for the corresponding simulation), which was compared to the same experimental data. As the 10 s stimulus data show a much faster decline after the peak than the 30 s stimulus data, both models fail to accurately capture the dynamics in Fig 3E. In the original paper for the vascular model, Barrett and co-workers speculate that this data discrepancy is due to few measurements on single vessels and that the fast signal decay in 10 s stimulus data would disappear if averaged over many vessels. Another confounding factor reported along with the experimental data ([46]) is that changes in the venous diameter of less than 2%, and capillary diameter changes of less than 7%, could not be captured using their measurement setup. For this reason, the data in Fig 3DE are at or below the detection limit, and we, therefore, view the disagreements with corresponding model simulations to not be a major concern.

In section 2.1.4, the model simulation fails to reach the first data point in Fig 5D (HbO, HbR, total Hb) and Fig 5E (HbR). This failure is due to the following inconsistency in the data. Examining the difference between the short and long OG inhibitory stimulation for the oxygenated hemoglobin, we would assume that the first data points up until t = 1 s would be approximately identical, as only one light pulse should have been emitted in both paradigms. However, for t = 1 s, experimental data points differ by a factor of two. As we assume that the effect of each light pulse is identical between short and long stimulation (since the model does not know the future), the model cannot describe this behavior. For this reason, we choose to keep the model with this shortcoming, since the alternative would be to have different parameters for short and long optogenetic stimulation.

In section 2.2.1, the model describes all data points except for the first data point in the LFP data for the positive response (Fig 6B, pink). In previous computational work based on the same data [36], parameters were allowed to change between stimulus and post-stimulus periods. This alternative approach gives a satisfactory agreement with data, including the first data point. However, we see no reason why the parameters should change when the stimulus ends,

and we, therefore, opt to accept the disagreement with the first data point. It is possible that a more non-linear description of LFP would provide a better agreement with this first data point. Again, we choose not to follow such a path, as it might lead to overfitting. A similar disagreement with experimental data can also be observed for the negative LFP response (Fig 6B, dark-purple). Here, the model simulations instantly return towards baseline upon stimulus cessation, while the experimental data show a ~ 1 s delay before rapidly changing. In Havlicek *et al.* [36], this was compensated for by simply prolonging the stimulation period by 1 s. We choose to follow the same path and alter the stimulation length to 21 s for the negative stimulation paradigm.

In section 2.3, the model prediction uncertainty for the positive response of CBF has high uncertainty, and therefore some parts of the prediction uncertainty lie outside of the data (Fig 7E, brown). Further, there might be dynamics in the changes of venous CBV for the inhibitory task, not captured by the model. However, because of the high uncertainty of the data, this discrepancy cannot be concluded (Fig 7C, blue).

### 3.4 Limitations of the current model

As with all models, there exists a set of limitations and assumptions which are necessary but still limit the scope of conclusions that can be drawn using the model.

First, our electrical activity description is simplified. More specifically, we use a simple phenomenological relationship between GABAergic interneurons and pyramidal neurons, where pyramidal neurons excite the GABAergic interneurons, which in turn act to inhibit the pyramidal neurons (Eq. 3). An important future development would be to integrate this model with common simulators of electrical activity, such as NEURON [74] or NEST [75] (see review in [76]). These models describe how membrane potential fluctuations arise as the result of the opening and closing of ion channels due to glutamatergic and GABAergic signaling. Such integration would also allow for the description of individual neurons, as there exist structurally and functionally different morphologies of interneurons (see [77] for review), which would be a more realistic description compared to our current lumped approach. One recent model [78] connects detailed electrophysiological models with control of SMCs around capillaries, which is an aspect of NVC control that is not included in our model.

Second, since we describe a wide variety of data, our choice of using the same model structure for all data has implied some options. For each experimental data set used in this study, we fitted some of the model parameters specific to each experimental study but kept all parameters the same within the same study, see Table 1. More specifically, all parameters are the same for the simulations within individual studies in Figs 3–4, 5, 6, 7 and S2, but some parameter values differ between the experimental setups within the studies (i.e., positive, and negative in the Shmuel study). The number of parameters that are replaced between the experimental setup correlates to how similar the setups are to each other. Most parameters are present over all studies (apart from the LFP scaling parameter which is only present in the Schmuel study), although $k_{u1}$, $k_{u2}$, and $k_{u3}$ have specific combinations in the Desjardins study. The parameter values of $k_{u1}$, $k_{u2}$, and $k_{u3}$ do change between the experimental setups within the Desjardins study. Details on parameter variations are given in S1 Appendix. Some other parameters are not fitted to any of the included data but are assumed to be the same across studies. For instance, the vascular volume fractions $v_{i,0}$ and baseline oxygen transport parameters (see section 4.2.4) are assumed to be the same across the different species. Furthermore, some experimental data measure individual arterioles and/or venules directly (Figs 3 and S2), while others measure a lumped estimate over the entire arterial or venous compartment (Figs 5–7). In the model, we use the same simulation variable to describe both individual and lumped measurements.

**Table 1. Overview of parameter usage over all experimental setups.**

| Study | Experimental setup | $k_{u1-3}$ | KPF, KIN, KINF | Sink$_{NO, NPY, PYR}$ | Intracellular signaling | K1-3, vis1-3 | Sign constants | ky4 (LFP) |
|---|---|---|---|---|---|---|---|---|
| Drew (Figs 3 and 4) | Sensory 125 ms | Y | Y | Y | Y | Y | | |
| | Sensory 10 s | | | | | | | |
| | Sensory 30 s | | | | | | | |
| Desjardins (Fig 5) | OG inhibitory | $k_{u1}$, $k_{u2}$ | Y | Y | Y | Y | | |
| | OG excitatory | $k_{u3}$ | | | | | | |
| | Sensory long | Unique | | | | | | |
| | Sensory short | Unique | | | | | | |
| Schmuel (Fig 6) | Positive | Unique | Unique | Unique | Y | Y | | Y |
| | Negative | Unique | Unique | Unique | | | Y | |
| Huber (Fig 7) | Positive | Unique | Y | Y | Y | Y | | |
| | Excitatory | Unique | | | | | | |
| | Negative | Unique | Y | Y | | | Y | |
| | Inhibitory | Unique | | | | | | |
| Drew (S2 Fig) | Sensory 125 ms | Y | Y | Y | Y | Y | | |
| | Sensory 10 s | | | | | | | |
| | Sensory 30 s | | | | | K2-3, vis2-3 | | |

The columns clarify the different groups of parameters, $k_{u1-3}$–stimulus scaling parameters, KPF, KIN, and KINF–neuronal signaling parameters, Sink$_{NO, NPY, PYR}$–elimination rate parameters, intracellular signaling parameters, K1-3, and vis1-3—viscoelasticity and stiffness coefficients, sign constants, and the LFP scaling parameter ky4. These groups of parameters are indicated where they are held the same over all experimental setups (rows) in a study and where they do differ. The studies are separated by a thicker line and by color. Within the studies, changes in parameters are indicated by altering color gradients within the color group and by notification explained below. Blank columns represent that the parameters are not included in the corresponding experimental setups.

Y–the same over all experimental setups within the study; Unique–parameters unique to this experimental setup in the study; OG–optogenetic.

Another highly simplified component of our model concerns metabolism. While the metabolism of oxygen occurs in the model, with both a baseline and a stimulus-dependent component, this part could, in principle, be expanded to incorporate the commonly measured metabolites in glia. These metabolites are typically measured using magnetic resonance spectroscopy (MRS), and models capable of describing these interplays already exist [79,80].

In addition, there are additional mechanisms of the NVC that we have not included. For instance, we do not include the role of astrocytes. Astrocytes have end-feet ideally placed around smooth muscle cells on arterioles and pericytes, and astrocytes are known to regulate the basal tone of arteries [81–84]. The reason we did not include these is that the exact role is unclear, and we did not find useful dynamic data. Another vasoactive pathway involves inward-rectifier $K^+$ ($K_{IR}2.1$) channels located on capillary endothelial cells, which open upon extracellular increases of $K^+$ as a result of neural activity [85]. The opening of these channels induces hyperpolarization, which propagates rapidly upstream to arteries [85]. We do not include direct actions on the capillaries, as reported in [86,87].

Lastly, in the model, the amount of some states (Eq. 3–7) is specified in arbitrary units, which means that we describe biological values such as concentrations, scaled by unknown scaling constants. Furthermore, the vascular states (Eq. 9–11) are normalized to a baseline value of 1, to improve computational stability and simplify calculations. This particular simplification is inherited from Barrett *et al.* [34,43,44]. Despite these renormalizations, our model can still be used to predict and simulate observations that are possible to capture in experimental data.

## 3.5 Strengths and potentials

The big potential of our model is not only the incorporation of more experimental data than any previous model (Fig 1C), but that we bypass uncertain parameter values and thus provide predictions with uncertainty, which can be useful in a variety of contexts. One of the main advantages of these predictions is that they can be used to understand mechanisms, which is illustrated in e.g., Fig 4. The simulations in Fig 4 are given by a single parameter vector, producing a single line. This is appropriate because the only question we ask is how the complex bimodal response can be produced by the model. For other types of questions, we want to plot predictions with uncertainty. Well-determined predictions, sometimes called core predictions [68,88], are useful for model validation and new experimental design [89]. Such validations are done for instance in Figs 5J or 7B, 7C and 7E (areas without solid lines). These core predictions can also be used to quantify non-measurable entities, which can be estimated from a combination of available data. One classical example of such model-based "measurements" is the approximate estimation of metabolism applying the Davis model [90] that uses the combination of BOLD and CBF data. In our comprehensive and complete model, a multitude of different data sets can be combined to estimate a variety of possible features. We, for instance, use our translational approach to create reasonable predictions for the role and timing of the three signaling arms (Fig 7F and 7G), the dynamics of hemoglobin (Fig 7H and 7I), and LFP (Fig 7J and 7K), even though such measurements are not available in humans. These estimated features could both aid the understanding of mechanisms in basic research and could be used to identify new biomarkers.

In other words, our new and integrative model could potentially integrate a wide variety of data to render the big picture of the neuronal state of a patient. This integrated understanding could potentially be used for diagnosis, stratifying between different patient groups, and treatment planning.

## 4. Methods

### 4.1 Model Formulation

The model is formulated using a mixture of ordinary differential equations (ODEs) and differential algebraic equations (DAEs). The general notation of such a model is as follows:

$$F(\dot{x}, x, z, \theta, u, t) = 0$$

$$F(\dot{x}(t_0, \theta), x(t_o, \theta), z(t_0, \theta), t_0) = 0$$

$$\hat{y} = g(x, z, \theta, u, t) = 0$$

in which $x$ is the vector of state variables for which derivatives are present; $z$ is the vector of state variables with no specified derivatives; $\dot{x}$ represent the derivatives of the states with respect to time $t$; $F$ and $g$ are non-linear smooth functions; $\theta$ is the vector of unknown constant parameters; where $u$ is the input signal corresponding to experimental data; where eq. 1b is the initial value solution, solving the values of the states at the initial time point $t = t_0$, which are dependent on the parameters $\theta$; and where $\hat{y}$ are the simulated model outputs corresponding to the measured experimental signals. Note that $x$, $z$, $\dot{x}$, $u$ and $\hat{y}$ depend on $t$, but that the notation is dropped unless explicitly needed.

### 4.2 Model structure

The model interaction graph is depicted in Fig 2. The interaction graph describes all reactions and interactions of the model. In practice, the presented model is the combination of multiple

previous studies, where section 4.2.1–4.2.3 builds upon the work of [42], 4.2.4–4.2.5 corresponds to the work of Barrett *et al.* [34,43,44], and finally, 4.2.6 corresponds to the work of Griffeth and Buxton [45]. The stimulus $u$ is given by the following equation:

$$u_i = \begin{cases} 1 \ t_{on} \leq t \leq t_{off} \\ 0 \ otherwise \end{cases}$$

where $t_{on}$, $t_{off}$ are the times when the signal goes on and off, respectively.

**4.2.1. Presynaptic activity and calcium influx.** Glutamate and GABA are released from different types of neurons upon membrane depolarization and bind to specific ion channel-coupled receptors located in the neuronal plasma membrane. Glutamate, an excitatory neuro-transmitter, binds to $\alpha$-amino-3-hydroxy-5-methyl-4-isoxazolepropionic acid (AMPA) and $N$-methyl-D-aspartate (NMDA) receptors. Activation of these ion channel-coupled receptors triggers ion-conducting pores to open, triggering an influx of $Na^+$ and $Ca^{2+}$ ions which cause a depolarization of the neuron. Depolarization opens voltage-gated calcium channels, allowing a further influx of $Ca^{2+}$ ions. In contrast, $\gamma$-aminobutyric acid (GABA) acts to prevent depolarization of neurons as it binds to the ion channel-coupled $GABA_A$ or $GABA_B$ receptor, which opens $Cl^-$ ion-conducting pores. Pyramidal neurons are glutamatergic, meaning that glutamate is released upon depolarization, and acts on *e.g.*, astrocytes and interneurons. GABAergic interneurons release GABA upon depolarization and target pyramidal neurons or other inter-neurons. This forms a simple relationship, where interneurons regulate the activity of pyramidal neurons. In the model, we represent these interplays between pyramidal and inhibitory interneuron activity and their combined effect on respective neuronal $Ca^{2+}$ levels as

$$\frac{dN_{NO}}{dt} = k_{u,NO} * sign_{u,NO} * u_1 + k_{PF1}E_{Pyr} - k_{IN1}E_{NPY} - sink_{N,NO}N_{NO}$$

$$\frac{dN_{NPY}}{dt} = k_{u,NPY} * sign_{u,NPY} * u_2 + k_{PF2}E_{Pyr} - k_{IN2}E_{NO} - sink_{N,NPY}N_{NPY}$$

$$\frac{dN_{Pyr}}{dt} = k_{u,Pyr} * sign_{u,Pyr} * u_3 + k_{INF1}N_{NO} - k_{INF2}N_{NPY} - sink_{N,Pyr}N_{Pyr}$$

$$\frac{d[Ca_i^{2+}]}{dt} = k_{Ca}(1 + N_i) - sink_{Ca,i}[Ca_i^{2+}]$$

$$E_i = \begin{cases} N_i \ N_i \geq 0 \\ 0 \ N_i < 0 \end{cases}$$

$$sign_u = \begin{cases} 1 \ if \ positive \ stimulation \\ decided \ by \ constants \end{cases}$$

Where $N_{NO}$, $N_{NPY}$, and $N_{Pyr}$ describe the neuronal activity of the GABAergic nitric oxide (NO) and neuropeptide Y (NPY) interneurons, and the pyramidal neurons, respectively; where $Ca_i^{2+}$ represents intracellular calcium in each respective neuron; $k_{u,i}$ is the input strength of the stimulation to the respective neuron; where $k_{PF1}$ and $k_{PF2}$ are the kinetic rate parameters governing pyramidal to GABAergic interneuron signaling; where $k_{INF1}$ and $k_{INF2}$ are the kinetic rate parameters describing the negative feedback from GABAergic interneurons to pyramidal neurons; where $k_{IN1}$ and $k_{IN2}$ are the kinetic rate parameters describing the negative

feedback between GABAergic interneurons; where $sink_{N,i}$ are kinetic rate parameters governing the degradation of the activity of respective neurons; where $k_{Ca}$ is the basal inflow rate of $Ca^{2+}$ ($k_{Ca} = 10$ for all three neurons), and finally, where $sink_{Ca,i}$ is the elimination rate of intracellular $Ca^{2+}$, and i = (Pyr, NO, NPY) denote the three different neuronal types.

**4.2.2. Pyramidal neuron signaling.** The rise in intracellular $Ca^{2+}$ levels in pyramidal neurons activates phospholipases which metabolize, through intermediary enzymatic steps, membrane phospholipids into intracellular arachidonic acid (AA) [12,69,91]. This is described by:

$$\frac{d[AA]}{dt} = k_{PL}\left[Ca_{Pyr}^{2+}\right] - \frac{k_{COX}[AA)}{K_{M,COX} + [AA]}$$

where $k_{PL}$ and $k_{COX}$ are kinetic rate parameters. In pyramidal neurons, AA is metabolized into prostaglandin E$_2$ (PGE$_2$) through a cyclooxygenase-2 (COX-2) and PGE synthase rate-limiting reaction [69,70]. PGE$_2$ evokes vasodilation through the activation of EP2 and EP4 receptors expressed on the surface of vascular smooth muscle cells (VSM) cells [69,92]. In the model, this mechanism described as

$$\frac{d[PGE_2]}{dt} = \frac{k_{COX}[AA]}{K_{m,COX+[AA]}} - k_{PGE_2}[PGE_2]$$

$$\frac{d[PGE_{2,vsm}]}{dt} = k_{PGE_2}[PGE_2] - sink_{PGE_2}[PGE_{2,vsm}]$$

Where PGE$_{2,vsm}$ represents PGE$_2$ acting on the VSM cells, and $k_{PGE2}$ and $sink_{PGE2}$ are kinetic rate parameters.

**4.2.3. GABAergic interneuron signaling.** The rise in intracellular $Ca^{2+}$ levels in GABAergic interneurons evokes the release of different vasoactive messengers and substances. For instance, NO has previously been shown to be a potent vasodilator at the level of arteries and arterioles in both *in vitro* and *in vivo* studies [87,92–94]. NO is released by specific NO-interneurons through a $Ca^{2+}$ dependent nitric oxide synthase (NOS) rate-limiting reaction. Here, we represent this process as

$$\frac{d[NO]}{dt} = k_{NOS}\left[Ca_{NO}^{2+}\right] - k_{NO}[NO]$$

$$\frac{d[NO_{vsm}]}{dt} = k_{NO}[NO] - sink_{NO}[NO_{vsm}]$$

where NO$_{vsm}$ represents NO acting on the VSM cells, and $k_{NOS}$, $k_{NO}$ and $sink_{NO}$ are kinetic rate parameters. Here, $sink_{NO} > 1\ s^{-1}$ following experimental work by [95].

Another potent vasoactive messenger is NPY. NPY is released by specific NPY producing subtypes of GABAergic interneurons and has previously been shown to induce vasoconstriction of vessels *in vitro* [92,96] and *in vivo* [47]. NPY binds to the NPY receptor Y1, a G$_{\alpha i}$-protein coupled receptor expressed on the surface of VSM cells enwrapping arteries and arterioles [97–99]. Activation of this receptor inhibits the synthesis of adenosine monophosphate (cAMP) and increases the intracellular $Ca^{2+}$ [100], leading to VSM contraction and

vasoconstriction. In the model, these mechanisms are represented as:

$$\frac{d[NPY]}{dt} = k_{NPY}\left[Ca_{NPY}^{2+}\right] - \frac{V_{max}[NPY]}{K_M + [NpY]}$$

$$\frac{d[NPY_{vsm}]}{dt} = \frac{V_{max}[NPY]}{K_M + [NpY]} - sink_{NPY}[NPY_{vsm}]$$

where NPY$_{vsm}$ represents NPY acting on the VSM cells, and $k_{NPY}$ and $sink_{NPY}$ are kinetic rate parameters. The conversion between intracellular NPY and NPY in the VSM, NPY$_{vsm}$, is governed by Michaelis-Menten kinetics [101,102], where $K_M$ is the Michaelis constant (the concentration of the substrate at which half max reaction rate is achieved) and V$_{max}$ is the maximal reaction rate.

The expression of the different vasoactive substances is scaled with a parameter, $ky_i$, $i =$ {*NO*, *PGE*2, *NPY*}, and summarized to a total vascular influence *G*:

$$G = k_{y,NO}([NO_{vsm}] - [NO_{vsm,0}]) + k_{y,PGE_2}([PGE_{2,vsm}] - [PGE_{2vsm,0}]) - k_{y,NPY}([NPY_{vsm}]$$
$$- [NPY_{vsm,0}])$$

### 4.2.4. Cerebrovascular dynamics described electrical circuit analog model by Barrett et al. (2012).

We describe the dynamics of three cerebrovascular compartments, corresponding to arteries/arterioles (*a*), capillaries (*c*), and veins/venules (*v*), respectively. The compartments are represented by an electrical circuit analogy, as originally presented in the work by Barrett *et al.* [34,43,44]. We use the derivative work of [103], and only present the fully derived equations to improve the clarity for the reader. We refer the reader to the original articles for an in-depth description of the equations [34,43,44,103]. The following equations describe the volume change for each respective cerebrovascular compartment, V$_i$, *with i =* {*a, c, v*}.

$$\frac{dV_a}{dt} = \frac{1}{k_{vis,a}}\left(\frac{K_a - \frac{V_a}{V_{a,0}}}{K_a - 1} + G - \frac{2}{C_{a,0}}\frac{V_a}{(r_1 + r_2)*f_1 + (r_2 + r_3)*f_{2+r_3f_3}}\right)$$

$$\frac{dV_c}{dt} = \frac{1}{k_{vis,c}}\left(\frac{K_c - \frac{V_c}{V_{c,0}}}{K_c - 1} - \frac{2}{C_{c,0}}\frac{V_c}{(r_2 + r_3)*f_{2+r_3f_3}}\right)$$

$$\frac{dV_v}{dt} = \frac{1}{k_{vis,v}}\left(\frac{K_v - \frac{V_v}{V_{v,0}}}{K_v - 1} - \frac{2}{C_{v,0}}\frac{V_v}{r_3f_3}\right)$$

Here, *K* are stiffness coefficients; $k_{vis,i}$ are viscoelastic parameters; $C_i$ represents the compliance of the vessel; *R* represents the vessel resistance; *f* represents the flow of blood between the cerebrovascular compartments; the baseline value is indicated by the subscript 0, and finally, *G* is the vasoactive function translating the actions of the vasoactive arms (Eq. 8) into hemodynamic changes.

In short, the arterial compartment is actively regulated by the vasoactive arms (Eq. 8 & 9a), and the evoked hemodynamic changes in the arterial vessels are propagated through capillary and venous compartments.

Using conservation of mass, the rate at which the blood volume change in a compartment is given by the difference between the in- and outflow of blood:

$$\frac{dV_i}{dt} = f_{i-1} - f_i$$

Restructuring the equation gives the following relationship:

$$f_i = f_{i-1} - \frac{dV_i}{dt}$$

Next, the pressure drop over a compartment is given by:

$$\Delta p_i(t) = \frac{1}{2} \sum_{i=1} r_i(f_{i-1} + f_i)$$

Which is subject to the pressure boundary conditions:

$$\sum_{i=1}^{3} \Delta p_i = \Delta p_r = 1$$

Using these conditions, we can solve for the inflow of blood into the first compartment $f_0$:

$$f_0 = \frac{2 - ((r_1 + r_2)f_1 + (r_2 + r_3)f_2 + r_3f_3)}{r_1}$$

The equations above form a DAE system, comprised of three volumes and four flows. To further simplify, the scales of the variables are normalized to:

$$\begin{bmatrix} \sum_{i=a}^{v} V_{i,0} = 1 \\ f_{i,0} = 1 \\ \sum_{i=a}^{v} r_{i,0} = 1 \end{bmatrix}$$

Using this, the rest of the physiological terms can be expressed as:

$$p_{i,0} = \begin{cases} \Delta p_r & if \ i = a \\ \Delta p_r - f_{i,0} \sum_{j=1}^{i-1} r_{j,0} & if \ i = \{c, v\} \end{cases}$$

$$c_{i,0} = \frac{V_{i,0}}{p_{i,0} - \frac{1}{2}r_{i,0}f_{i,0}}$$

$$l_i = (r_{i,0}v_{i,0}^2)^{\frac{1}{3}}$$

$$r_i = \frac{l_i^3}{v_i^2}$$

where $P$ is the pressure at the entry point to each compartment, $C$ is the compliance of the vascular vessel, $L$ is the length of the vascular segment, and $R$ is the vessel resistance to flow.

Given Eqs 9–16, the initial conditions for the cerebrovascular volumes $V_i$ and the vascular resistance $R_i$ must be specified. We set these initial conditions in accordance with the work of Barrett *et al.* [34], which based these values on the available literature, resulting in $V_{i,0}$ = [0.29, 0.44, 0.27] and $R_{i,0}$ = [0.74, 0.08, 0.18].

**4.2.5. Oxygen transportation.**   To describe the oxygen transportation through the cerebrovascular system, we describe three vascular compartments and a cerebral-tissue compartment. The amount of oxygen in these compartments is given by:

$$\frac{dn_{O_2,a}}{dt} = f_0 C_{O_2,in} - f_1 C_{O_2,a,c} - j_{O_2,a} - j_{O_2,s}$$

$$\frac{dn_{O_2,c}}{dt} = f_1 C_{O_2,a,c} - f_2 C_{O_2,c,v} - j_{O_2,c}$$

$$\frac{dn_{O_2,v}}{dt} = f_2 C_{O_2,c,v} - f_3 C_{O_2,out} - j_{O_2,v} + j_{O_2,s}$$

$$\frac{dn_{O_2,t}}{dt} = \sum_i j_{O_2,i} - CMRO_2$$

where the flow of blood $f$ transports oxygen in and out from each compartment. The permeability in the vessel wall allows for oxygen to diffuse to the tissue compartment, $jO2_i$, where $i = \{a, c, v\}$. An arterio-venous diffusion shunt is described by $jO2_s$, where oxygen moves from the arteriole to the venous compartment. Oxygen is metabolized in the tissue compartment, $CMRO_2$. The metabolism of oxygen is given by:

$$CMRO_2 = CMRO_{2,0}(1 + k_{met}(N_{NO} + N_{NPY} + N_{Pyr}))$$

where $CMRO_{2,0}$ is the basal metabolism of oxygen, which is increased by the activity level of each neuron (see Eq. 3) scaled by the parameter $k_{met}$.

The concentration of oxygen, $C_{O_2,i}$ is given by the amount of oxygen, $n_{O_2,i}$, divided by the volume of each compartment, $V_i$, except for the oxygen concentration that enters the arteries, $c_{O_2,0}$, which is assumed to be constant, minus a loss of oxygen to the surrounding tissue $c_{O_2,leak}$.

$$C_{O_2,i,i+1} = \begin{cases} \dfrac{n_{O_2,i}}{V_i} & if \ i = \{a, c, v, t\} \\ C_{O_2,0} - C_{O_2,leak} & if \ i = in \end{cases}$$

Where the tissue volume fraction $V_t$ = 34.8 [43] and $c_{O_2,leak} = 0.116 \ mM$ (calculated using data from [43,104]).

As in the original work, by ignoring the minor fraction of oxygen directly dissolved in the blood plasma, blood oxygen concentration can be related to oxygen partial pressures through the oxygen-hemoglobin saturation curve:

$$p_{O_2,i,i+1} = p_{50} \left( \frac{C_{O_2,max}}{C_{O_2,i,i+1}} - 1 \right)^{-\frac{1}{h}}$$

Where $p_{50}$ = 36 $mmHg$ is the oxygen partial pressure at the halfway saturation point, $cc_{O_2,max}$ = 9.26 $mM$ is the maximum oxygen concentration in blood, and $h$ = 2.6 is the Hill exponent. These values are taken from [105].

The oxygen pressure in the tissue, $p_{O_2,t}$, is calculated using Henry's law:

$$p_{O_2,t} = \frac{C_{O_2,t}}{\sigma_{oxygen}}$$

Where $\sigma_{Oxygen}$ = 1.46 $\mu M/mmHg$ is the coefficient for solubility of oxygen in tissue [106]. The average pressure, $\bar{p}_{O_2,i}$, in a compartment is obtained by averaging over the input and output pressures:

$$\tilde{p}_{O_2,i} = \frac{p_{O_2,i-1} + p_{O_2,i+1}}{2}$$

The diffusion of oxygen between the vessels and the tissue, $j_{O_2,i}$, as well as between the arteries and the veins, $j_{O_2,s}$, is driven by the difference in partial oxygen pressure:

$$j_{O_2,i} = g_i(\tilde{p}_{O_2,i} - p_{O_2,t}), i = \{a, c, v\}$$

$$j_{O_2,s} = g_s(\tilde{p}_{O_2,a} - \tilde{p}_{O_2,v})$$

where $g_i$, $i = \{a, c, v, s\}$ are rate constants.

We employed the identical strategy of Barrett and Suresh [43] (see supplementary material in [43]), and used the experimental $pO_2$ data from Vovenko [104] to calculate $g_i$, $c_{O_2,leak}$, and $CMRO_{2,0}$. This leaves the parameter $k_{met}$ to be estimated where applicable.

The oxygen saturation of blood in the different vascular compartments, $S_iO_2$, is approximated by dividing the average oxygen concentration of each compartment with $c_{O_2,max}$:

$$S_iO_2 = \frac{C_{O_2,i-1,i} - C_{O_2,i,i+1}}{C_{O_2,max}}, i = \{a, c, v\}$$

Lastly, using the blood oxygen saturation $S_iO_2$ and vascular volumes ($V_i$), corresponding changes in oxygenated hemoglobin (HbO), deoxyhemoglobin (HbR), and total hemoglobin (HbT) are given by:

$$n_{Hbt,i} = V_i$$

$$n_{HbO,i} = V_i S_i O_2$$

$$n_{HbR,i} = V_i(1 - S_i O_2)$$

**4.2.6. BOLD signal derivation.** Finally, the oxygen saturation of blood in the different vascular compartments, $S_iO_2$ (Eq. 24) and the vascular volumes, $V_i$ (Eq. 9) are used to calculate the BOLD signal. The following derivation of the BOLD signal equations and parameter values are taken from the work of [45], and we refer the reader to the original article for a detailed view of the equations. The BOLD signal is expressed as a summation of the contributions from each vascular compartment $\{a, c, v\}$ and an extracellular compartment $\{e\}$ symbolizing the

tissue:

$$BOLD = H\sum_j S_j, \; j = \{a, c, v, e\}$$

$$H = \frac{1}{V_e + \sum_k \varepsilon_k \hat{V}_k}, k = \{a, c, v\}$$

$$\varepsilon_{(a,c,v)} = \frac{S_{(a,c,v),0}}{S_{e,0}} = \lambda \frac{e^{-TER^*_{2(a,c,v),0}}}{e^{-TER^*_{2e,0}}}$$

Where $S_j$ is the intrinsic signal from each compartment; $V_e = 1 - \hat{V}_k$ is the extravascular volume; $\hat{V}_k = V_{iv,0} \sum V_k$ is the intravascular volume scaled by the baseline intravascular volume fraction ($V_{iv,0} = 0.05$, [107]); $\varepsilon_i$ is the intrinsic signal ratio of blood and $\lambda$ is the intravascular to extravascular spin density ratio (we assume $\lambda = 1.15$); $R^*_2$ is the transverse signal relaxation rate for the four compartments, and $TE$ is the echo time used during image acquisition in each study.

The transverse signal relaxation rate for the extravascular compartment ($R^*_{2e,0}$) was assumed to be $25.1s^{-1}$ [108], and was calculated for the intravascular compartments by utilizing the following quadratic expression taken from [109]:

$$R^*_{2i,0} = A^* + C^*(1 - S_iO_2)^2$$

where

$$A^* = 14.87Hct + 14.686$$
$$C^* = 302.06Hct + 41.83$$

These equations are governed by $S_iO2$ (Eq. 24) and the resting hematocrit in blood ($Hct$; $Hct_{a,v} = 0.44$, $Hct_c = 0.33$ [110,111]).

The respective signal contribution from each compartment is given by:

$$S_i = \begin{cases} \varepsilon_i \hat{V}_i e^{-TE\Delta R^*_{2i}} \; if \; i = \{a, c, v\} \\ V_e e^{-TE\Delta R^*_{2e}} \; if \; i = e \end{cases}$$

Where $\Delta R^*_2$ (*i.e.*, the change in MR signal relaxation rate) is given by the following expressions:

$$\Delta R^*_{2i} = C^*((1 - S_iO_2)^2 - (1 - S_iO_{2,0})^2)$$

$$\Delta R^*_{2e(a,c,v)} = \begin{cases} \frac{4\pi}{3}\Delta_x Hct\gamma B_0 \sum_{i=a,v}[\hat{V}_i(|S_{off}O_2 - S_iO_2|) - \hat{V}_{i,0}(|S_{off}O_2 - S_iO_{2,0}|)] \\ 0.04(\Delta_x Hct\gamma B_0)^2[\hat{V}_c(|S_{off}O_2 - S_cO_2|)^2 - \hat{V}_{c,0}(|S_{off}O_2 - S_cO_{2,0}|)] \end{cases}$$

Here, $\Delta\chi = 2.64\times10^{-7}$ is the susceptibility of fully deoxygenated blood [112]; $\gamma = 2.68\times10^8$ is the gyromagnetic ratio of protons; $B_0$ is the magnetic field strength used during image acquisition, and finally, $S_{off}O_2 = 0.95$ is the blood saturation which gives no magnetic susceptibility difference between blood and tissue [112].

**4.2.7. LFP simulations.**   Finally, for the fitting to the LFP data in macaques, we use the following expression:

$$LFP = k_{y4}N_{Pyr}$$

where $N_{Pyr}$ is the phenomenological activity state of pyramidal neurons, and $k_{y_4}$ is an unknown scaling constant.

## 4.3. Model evaluation

**4.3.1 Optimization of parameters.**   Once the model structure has been formulated (Fig 2) and data has been collected (data acquisition described below), the parameters, $\theta$, need to be determined. This is commonly done by evaluating the negative log-likelihood function. Assuming independent, normally distributed additive measurement noise, the negative logarithm of the likelihood of observing data $D$ given $\theta$ is:

$$J(\theta) = -\log(L(D|\theta)) = \frac{1}{2}\sum_{e=1}^{n_e}\sum_{o=1}^{n_o^e}\sum_{s=1}^{n_s^{e,o}}\left[\log(2\pi(\sigma_s^{e,o})^2) + \left(\frac{y_s^{e,o} - \hat{y}_s^{e,o}(\theta)}{\sigma_s^{e,o}}\right)^2\right]$$

where $n_e$ is the number of experiments $e$; where $n_o^e$ is the number of observables $o$ per $e$; where $n_s^{e,o}$ is the number of samples per $o$ and $e$; where $\sigma_s^{e,o}$ is the standard deviation of the data point; where $y_s^{e,o}$ is the measured data point; where $\hat{y}(\theta)$ is the corresponding simulated data point. By maximizing $L$, the maximum likelihood estimate of the unknown parameters, $\theta$ can be obtained. However, it is more common and more numerically efficient to minimize the equivalent negative log-likelihood function. If the measurement noise $\sigma_s^{e,o}$ is known, $J(\theta)$ share the same optimal parameters with the least-squares function $J_{lsq}$:

$$J_{lsq}(\theta) = \sum_{e=1}^{n_e}\sum_{o=1}^{n_o^e}\sum_{s=1}^{n_s^{e,o}}\left(\frac{y_s^{e,o} - \hat{y}_s^{e,o}(\theta)}{\sigma_s^{e,o}}\right)^2$$

In addition to this quantitative comparison with data, qualitative insights are added to the function to be optimized, converting $J_{lsq}(\theta)$ to $J_{opt}(\theta)$ (see below). In practice, the parameters are determined by optimizing $J_{opt}(\theta)$ over $\theta$ by using various optimization algorithms (see Section 4.5 below), i.e. they are given by:

$$\hat{\theta} = \arg\min J(\theta)$$

where $\hat{\theta}$ are the optimal parameters.

**4.3.2 Simulation uncertainty.**   The model uncertainty was estimated using a Markov chain Monte Carlo (MCMC)-sampling procedure with $10^5$ samples (see section 4.5 for details), generating posterior distributions of the parameters, and collecting all $\chi^2$ acceptable parameters encountered. Using the acceptable $\chi^2$ parameters, a confidence interval:

$$CI_{\alpha,df} = (J_{lsq}(\theta) \le J_{lsq}(\hat{\theta}) + \Delta_\alpha(X_{df}^2))$$

was drawn where the threshold $\Delta_\alpha(\chi_{df}^2)$ is the $\alpha$ quantile of the $\chi_{df}^2$ statistics [113,114]. We also require that all acceptable parameters should pass a $\chi^2$-test.

For a small fraction of the parameters identified using the optimization approach described above, certain types of unrealistic behavior may appear. For instance, in Fig 5, regarding the Desjardins mice data, less than 1% of the parameters display a rapid spike after stimuli has ceased. Such unrealistic spikes are not prohibited by the cost function because the simulation

has returned to the next data point. However, we judge such rapid spikes to be unrealistic and thus remove them from the uncertainties.

**4.3.3 Preservation of qualitative features between the species and experiments.**   The optimization is done using a cost function that minimizes the quantitative agreement between each dataset studied, while also fulfilling the qualitative demands and mechanistic insights obtained from the analysis of other datasets (Section 4.3.1). This is done by the addition of extra terms to the cost function, which are zero if the qualitative demand is fulfilled, and high otherwise. On a top-level description, those qualitative demands are as follows:

- *Insights from Figs 3 and 4*: The vasoactive substance NO, produced by the NO-interneurons, has initially fast dynamics compared to the $PGE_2$-release by the pyramidal cells. After this fast initial response, the vasoactive control by the NO-interneurons is quickly turned off again, leaving the pyramidal cells as the primary dilator for prolonged stimulations, sometimes causing a two-peak response. The post-peak undershoot is caused by the slow NPY-release caused by the NPY-interneurons, which is vasoconstricting also after the vasodilating controls have subsided.

- *Insight from Fig 5*: The oxygenated and deoxygenated hemoglobin responses should have the same timing, and the difference in amplitude between these two responses should not be too big.

- *Insight from Fig 6*: Following a stimulation, a fast peak in LFP is produced, which thereafter diminishes down to a lower plateau, where LFP remains for the rest of the stimulation. After the stimulation has been turned off, LFP quickly declines to values below baseline, whereafter it recovers back to the baseline again. For a negative BOLD response, this behavior in the LFP is inverted.

Quantitative specifications of these qualitative demands are given in S2 Appendix, Eqs S1–15, and all optimization scripts are provided, together with the analyzed models and data.

**4.3.3 Handling differences between negative and positive BOLD response.**   For the data with both a positive and a negative BOLD (Figs 6 and 7), we allow the input stimulus parameters ($k_{u,i}$) to change. Furthermore, we also allow the parameters governing the inter-neuronal interactions ($k_{PF1}$, $k_{PF2}$, $k_{INF1}$, $k_{INF2}$, $k_{IN1}$, $k_{IN2}$), and the parameters for neuronal recovery ($sink_{N,NO}$, $sink_{N,NPY}$, $sink_{N,Pyr}$) to change between positive and negative responses. These differences are allowed for, and implemented, due to a similar need as was done in our previous model for positive and negative BOLD [39]. The main limitation of this model regarding negative BOLD is that such dynamics also involve neuronal crosstalk and inhibition between different brain areas, which only is described in a highly simplified way in our models.

## 4.4 Experimental data

In this work, experimental data from a total of five different studies have been used, which includes data from rodents, primates, and humans. Below, condensed descriptions of the experimental data are provided, and we kindly refer the reader to the original manuscripts for a complete description of the experimental methods. All data were extracted from figures in the respective original publications using the tool WebPlotDigitizer (https://automeris.io/WebPlotDigitizer/).

**4.4.1 Fractional diameter change of arterioles and venules in mice.**   The work of Drew *et al.* [46] presents experimental data of fractional diameter change in arterioles and venules upon different lengths of sensory stimulation in mice (n = 21), measured using 2-photon imaging. The sensory stimulation consisted of trains of air puffs (delivered at 8 Hz, 20 ms

duration per puff) directed at the vibrissae, with three different durations: 20 ms, 10 s, and 30 s. The mice were awake during the experiment but were briefly anesthetized before the imaging session using isoflurane (2% in air). We have chosen to consider the period after each puff as a part of the stimulation cycle. This makes the shortest stimulation length of one air puff 125 ms long. The vascular responses were extracted from Fig 2C in the original publication [46].

**4.4.2 Arteriolar diameter responses for optogenetic and sensory stimuli in mice.** Uhlirova *et al*. [47] report arteriolar diameter responses to both optogenetic and sensory stimulations measured using *in vivo* two-photon imaging, during both awake and under anesthesia conditions in mice. Briefly, four different stimulation protocols were used in their study: three conducted under the administration of the anesthetic α-chloralose, and the fourth during the awake condition. Below, summarized descriptions of each stimulation protocol are presented, and we refer the reader to the original publication [47] and our previous modeling paper [42] for a complete description.

### Optogenetic stimulation of pyramidal neurons during anesthesia

Thy1-ChR2-YFP mice (n = 2 subjects), with Channelrhopodsin-2 (ChR2), selectively expressed in layer 5 pyramidal neurons, were imaged (28 measurement locations along 13 arterioles within 40–380 μm range) during optogenetic stimulation (single light pulse lasting 50–80 ms) under both control conditions and under the influence of both the AMPA/kainate receptor antagonist 6-cyano-7-nitroquinoxaline-2,3-dione (CNQX), and the NMDA receptor-selective antagonist D-(-)-2-Amino-5-phosphonopentanoic acid (AP5).

### Optogenetic stimulation of GABAergic interneurons during anesthesia

VGAT-ChR2(H134R)-EYFP mice (n = 3 subjects), with ChR2 selectively expressed in GABA interneurons, were imaged (52 measurement locations along 25 arterioles within 50–590 μm range) during optogenetic stimulation (one pair of light pulses separated by 130 ms, for a total of 450 ms duration) under both control conditions (Fig 5D, black error bars) and during application of the NPY receptor Y1 inhibitor BIBP-3226.

### Sensory stimulation during anesthesia

Additionally, four wild-type mice underwent imaging (10 measurement locations along 7 arterioles within 130–490 μm range) during sensory stimulation (2 s train of electrical pulses) under both control conditions and during application of the NPY receptor Y1 inhibitor BIBP-3226.

### Sensory and Optogenetic stimulation during the awake condition

Lastly, four wild-type and three VGAT-ChR2(H134R)-EYFP awake mice were imaged during sensory stimulation (1 s of air puffs, 100 ms puffs at 3 Hz) and optogenetic stimulation (single light pulse lasting 150–400 ms), respectively.

For the data extracted from this study, the standard error of the mean (SEM) in all extracted time series with SEM measurements smaller than the mean SEM was set to the mean SEM. This was done to avoid overfitting the model to a few extreme data points.

**4.4.3 Hemoglobin and BOLD responses for optogenetic and sensory stimuli in mice.** Desjardins *et al*. [48] present experimental data consisting of optical intrinsic imaging (OIS) of blood oxygenation for both wild-type mice (n = 16) and two mice-lines transfected with (ChR2). The mice were kept in an awake state during the data acquisition. In the transfected mice, ChR2 was expressed in pyramidal neurons (Emx1-Cre/Ai32) or all inhibitory

interneurons (VGAT-ChR2(H134R)-EYFP). For the wild-type mice, a sensory stimulus consisting of air puffs (3–5 Hz) delivered to a whisker pad (duration 2 or 20 s) was used to induce hemodynamic responses. The optogenetic-transfected mice were exposed to an optogenetic stimulus consisting of a single 100 ms light pulse (473 nm) or a 20 s block of 100 ms light pulses delivered at 1 Hz. The blood oxygenation data were analyzed using the baseline assumption of 60 $\mu M$ oxygenated hemoglobin and 40 $\mu M$ deoxygenated hemoglobin. In addition, some Emx1-Cre/Ai32 mice underwent BOLD fMRI imaging for the 20 s optogenetic stimulus using gradient-echo echo-planar-imaging (GE-EPI) readout (TE/TR = 11–20 ms / 1 s).

**4.4.4 Electrophysiological and BOLD responses in macaque monkeys.**   The study by Shmuel *et al.* [10] report simultaneously measured electrophysiological responses by invasive recording, and non-invasively measured BOLD responses. These measurements were carried out in the visual cortex of anesthetized macaques at 4.7 T (n = 7). The stimulus paradigm consisted of blocks of 20 s visual stimulus featuring flickering radial checkers rotating at 60˚/s. A grey background persisted for 5 s before and 25 s after the stimuli. If the visual stimuli overlapped with the receptive field of V1, it induced positive BOLD responses in close vicinity to the electrode. In contrast, if the visual stimuli were presented outside the receptive field of V1, it induced negative BOLD responses in the same area. The electrophysiological recordings were processed by averaging the fractional change of the power spectrum over the frequency range of 4–3000 Hz, using a temporal resolution of 1 s. BOLD data were acquired using a gradient-echo echo-planar-imaging (GE-EPI) readout (TE = 20 ms, TR = 1 s, in-plane spatial resolution 0.75 x 0.75 x 2 mm$^3$). Positive and negative BOLD responses evoked by the two visual stimuli were sampled and averaged over the same voxels located around the electrode. These processed responses were extracted from Figs 1D (BOLD) and 2A (electrophysiological responses) in the original publication [10], respectively. The SEM in all extracted LFP time series with SEM measurements smaller than the mean SEM was set to the mean SEM, to avoid overfitting to a few extreme data points.

**4.4.5 MR-based hemodynamic measurements in humans.**   The study by Huber *et al.* [49] presents *in vivo* MR-based measurements of CBF, CBV, and BOLD changes in humans (n = 17) evoked using multiple visual stimuli. The study used three different visual stimulation paradigms (alternating 30 s rest vs. 30 s stimulation), consisting of two different full-field flickering checkerboards and one small circle flickering checkerboard paradigm. The full-field flickering checkerboard paradigms were used to measure arterial CBV, venous CBV, and total CBV data for both an excitatory and an inhibitory task. The small circle paradigm was used to evoke strong negative BOLD responses, and to concurrently measure both positive and negative BOLD responses, as well as the associated total CBV changes. These data were acquired using a Slice-Saturation Slab-Inversion Vascular Space Occupancy (SS-SI-VASO) sequence [115] with a multi-gradient echo EPI readout (*TE$_1$/TE$_2$/TE$_3$/TI$_1$/TI$_2$/TR* = 12/32/52/1000/2000/2500/3000 ms, nominal resolution = ~1.3 x ~1.3 x 1.5 mm$^3$). In addition, CBF changes were acquired from ROIs of positive and negative BOLD responses using a pulsed arterial spin label (PASL) sequence in four subjects (*TE$_1$/TE$_2$/TI$_1$/TI$_2$/TR* = 8.2/19.4/700/1700/3000 ms, nominal resolution = 3 x 3 x 3 mm$^3$). BOLD and total CBV responses were extracted from Fig 4A and 4B; arterial CBV, venous CBV, and total CBV responses were extracted from Fig 5B and 5C, and finally, CBF responses were extracted from Fig 6C in the original publication.

## 4.5. Model implementation

Implementation and simulation of the model equations (specified in 2.2) were carried out using the 'Advanced Multi-language Interface to CVODES and IDAS' (AMICI) toolbox [116,117]. Parameter estimation for the model evaluation was carried out using the

MATLAB implemented 'MEtaheuristics for systems biology and bioinformatics Global Optimization' (MEIGO) [118] toolbox and the enhanced scatter search (eSS) algorithm. In addition, eSS was used in conjunction with two local optimization solvers: dynamic hill climbing [119] and the interior point algorithm included in FMINCON (MATLAB and Optimization Toolbox Release 2017b, The MathWorks, inc., Natick, Massachusetts, United States) paired with the calculation of the objective function gradient using AMICI. For the model uncertainty analysis, we employed the 'Parameter EStimation ToolBox' (PESTO) [120] and the region-based adaptive parallel tempering algorithm [121] to generate the posterior distributions of the parameters. The parameter bounds were [-4.5,4.5] in $\log_{10}$ space unless specified otherwise in Section 4.2.

## Supporting information

**S1 Appendix. Posterior probability profiles of the model parameters and model estimation.**
(DOCX)

**S2 Appendix. Qualitative demands during simulation.**
(DOCX)

**S1 Fig. Parameter probability profile to Fig 3.** Posterior probability profile (y-axis) for each estimated model parameter (x-axis, $\log_{10}$ space) for the model estimated to data presented by Drew et al. [46].
(TIF)

**S2 Fig. Model estimation of Fig 3 with different viscoelasticity parameters.** Model estimation to experimental data of arteriolar (A–C) and venular (D–F) volume changes in awake mice for three different sensory stimulation lengths: 125 ms (A & D), 10 s (B & E), and 30 s (C & F). The viscoelasticity and stiffness coefficients of the capillary and venous compartment change at t = 2s for the long stimulation (C & F). Experimental data are replotted versions of data presented in Fig 2C of the original manuscript [46]. The stimulation lengths are denoted with the black bar in the bottom left portion of each graph. For each graph: experimental data (colored symbols); The uncertainty of the experimental data is presented as SEM (colored error bars); the best model simulation is seen as a colored solid line; the model uncertainty as colored semi-transparent overlays. The x-axis represents time in seconds, and the y-axis is the normalized vessel diameter change ($\Delta$%).
(TIF)

**S3 Fig. Parameter probability profile to S2 Fig.** Posterior probability profile (y-axis) for each estimated model parameter (x-axis, $\log_{10}$ space) for the model estimated to data presented by Drew et al. [46], allowing the viscoelasticity and stiffness coefficients of the capillary and venous compartment change between 10 and 30 s stimulation.
(TIF)

**S4 Fig. Model evaluation and model prediction of drug perturbations to arteriole response data from awake and anesthetized mice.** A-E: Model evaluation to arteriolar response data from awake (A, B) and anesthetized animals (C–E) for both optogenetic (OG) (B, D, E) and sensory (A, C) stimuli. F-H: Model predictions to drug perturbed arteriolar response data during anesthesia condition. OG stimulation (G, H) and sensory stimulation (F), during the presence of the NPY receptor Y1 antagonist BIBP (F, G) and glutamatergic signaling blockers AP5 and CNQX (H). For each graph, the best-estimated model simulation (solid red line, A-E only) paired with model uncertainty (red shaded areas) compared to experimental data (black

symbols, error bars depicting standard error of the mean). The stimulation length is indicated by the black bar in the lower left portion of each graph. Experimental data originates from Uhlirova et al. [47].
(TIF)

**S5 Fig. Parameter probability profile to Fig 5.** Posterior probability profile (y-axis) for each estimated model parameter (x-axis, $\log_{10}$ space) for the model estimated to data presented by Desjardins et al. [48]. The affix S represents the parameter value for sensory stimulation and is further subdivided into short and long sensory stimulation.
(TIF)

**S6 Fig. Parameter probability profile to Fig 6.** Posterior probability profile (y-axis) for each estimated model parameter (x-axis, $\log_{10}$ space) for the model estimated to data presented by Shmuel et al. [10]. The affix neg represents the parameter values for the negative response.
(TIF)

**S7 Fig. Parameter probability profile to Fig 7.** Posterior probability profile (y-axis) for each estimated model parameter (x-axis, log10 space) for the model estimated to data presented by Huber et al. [49]. The affix neg represents the parameter values for the negative response, exc represents the parameter values for the excitatory response, and inh represents the parameter values for the inhibitory response.
(TIF)

**S1 Table. Limits of the parameter posterior for Figs 3 and S1.**
(XLSX)

**S2 Table. Limits of the parameter posterior for S2 Fig.**
(XLSX)

**S3 Table. Limits of the parameter posterior for Figs 5 and S5.**
(XLSX)

**S4 Table. Limits of the parameter posterior for Figs 6 and S6.**
(XLSX)

**S5 Table. Limits of the parameter posterior for Figs 7 and S7.**
(XLSX)

## Author Contributions

**Conceptualization:** Sebastian Sten, Gunnar Cedersund.

**Formal analysis:** Sebastian Sten, Henrik Podéus, Nicolas Sundqvist.

**Funding acquisition:** Maria Engström, Gunnar Cedersund.

**Investigation:** Sebastian Sten, Henrik Podéus.

**Methodology:** Sebastian Sten, Henrik Podéus.

**Project administration:** Sebastian Sten, Maria Engström, Gunnar Cedersund.

**Supervision:** Fredrik Elinder, Maria Engström, Gunnar Cedersund.

**Validation:** Sebastian Sten, Henrik Podéus, Nicolas Sundqvist.

**Visualization:** Sebastian Sten, Henrik Podéus, Nicolas Sundqvist.

**Writing – original draft:** Sebastian Sten, Henrik Podéus, Gunnar Cedersund.

**Writing – review & editing:** Sebastian Sten, Henrik Podéus, Nicolas Sundqvist, Fredrik Elinder, Maria Engström, Gunnar Cedersund.

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
