## [Decision Letter · Decision Letter 0]

24 Aug 2022

Dear Dr. Cedersund,

Thank you very much for submitting your manuscript "A quantitative model for human neurovascular coupling with translated mechanisms from animals" for consideration at PLOS Computational Biology.

As with all papers reviewed by the journal, your manuscript was reviewed by members of the editorial board and by several independent reviewers. In light of the reviews (below this email), we would like to invite the resubmission of a significantly-revised version that takes into account the reviewers' comments.

We cannot make any decision about publication until we have seen the revised manuscript and your response to the reviewers' comments. Your revised manuscript is also likely to be sent to reviewers for further evaluation.

Sincerely,

Alison L. Marsden

Academic Editor

PLOS Computational Biology

Jason Haugh

Section Editor

PLOS Computational Biology

Reviewer's Responses to Questions

**Comments to the Authors:**

Reviewer #1: This is a very interesting and comprehensive paper on the neurovascular coupling, the means by which blood flow is regulated throughout the brain. A mechanistic understanding of the NVC is important because functional neuroimaging uses hemodynamic changes as a proxy for neural activity.

In general, I think this paper is a good fit for PLOS Computational Biology, and it seems to be innovative. However, I found several aspects of the paper very unclear and would suggest clarification before publication.

1. I believe that each of the levels (neural response, hemodynamics, and BOLD response) are decoupled - that is, for instance, that the neural response affects the hemodynamics but not vice versa. If this is the case, that should be clarified; Fig. 2 is trying to mimic anatomy in some sense and is not depicting the modeling accurately. Additionally, in several of the schematics, there are arrows pointing to nothing, which makes the interactions unclear.

2. In the discussion, the authors mention that no existing model can describe "pharmacological perturbations including the effect of anesthetics", followed by a sentence that implies that this model can describe all of the aforementioned behaviors simultaneously. I did not see any information on anesthetics, so the statement seems misleading to me. This should be clarified.

3. The second limitation mentions the parameters used for different experiments and attempts to explain which were preserved or held constant and which were changed where. It is extremely unclear, and not helped by the fact that all the parameters are only shown in the supplementary information. I believe this could be conveyed more clearly with a table or perhaps a figure.

4. The readability of the paper was severely impacted by embedding figure captions but not figures. As PLOS's guidelines for submissions allow this, please include a copy that has figures and captions in the text.

5. The timing described in lines 234-236 is very confusing.

6. The paper includes too many acronyms and uses them inconsistently - for instance, optogenetics is (rightfully!) used most of the time in the text, and only used as an abbreviation in figure. Please consider replacing some acronyms (CBF, CBV, OG, etc.) with the full words in the text. Also, MCMC is used without definition.

7. Relatedly: define all acronyms in figures. For instance, in Fig. 2, I believe u, f_i, and V_i are not defined.

8. The figures would benefit from some careful thought about design. A few examples - the "brown" in line 208 looks like red. Fig. 1C would be easier to understand if colors were used in addition to text. The legend in Fig. 7 is not even in order for each subplot. Also I think the same colors are used for CBV and CBF.

Reviewer #2: The authors develop a model to couple i) neuron activity as a set of ODEs, ii) release of vasodilation/vasoconstriction molecules downstream of neural activity as a set of ODEs representing relevant reactions, iii) blood flow from a lumped model akin to electrical circuits, iv) oxygen metabolism downstream of blood flow. The model builds off previous work by connecting the different model modules from prior work. To calibrate the model the authors follow a Bayesian approach to explain a comprehensive dataset. The calibration of the model provides insights into the unobserved model signaling dynamics from the observed blood flow and oxygen observed data. While the model setup is comprehensive and data from the literature is also comprehensive, I do have some major comments:

[1] Line 185: cross reference to validation data seems wrong. The same cross reference is repeated later in the text. Please check other cross references in case I missed other ones, but the validation ones are crucial which is why I noticed this.

[2] Line 233: How was the stimulus prescribed in the model?

[3] Line 242: where all the model parameters fitted? The authors seem to be using a Bayesian approach which makes me think that distributions of the parameter posterior are estimated. That would be great, much better than standard fitting, but I am definitely curious how loose/tight were the priors and whether the posteriors are reasonable. The authors should refer the reader to these statistics in this section, even if the statistics of the Bayesian fitting are reported in the Supplement. I understand some of the priors are defined in the methods, but a cross reference to the posterior traces in the Results would be great.

[4] Figure 4: What are u_1, u_2, u_3? Are these the inputs to the simulation? This is related to my comment [2], it is not clear what drives the simulation

[5] Figure 4 E-G, why are NO values negative? I understand these are arbitrary units but in principle the three variables shown in these plots are concentrations and are expected to be positive. Since the Methods is left for later on in the manuscript, a note as to why we can expect negative values for these ‘concentrations’ should be added.

[6] Line 348: Were the input parameters manually adjusted or all parameters, including inputs, determined with a Bayesian calibration? And if the inputs were part of the optimization, were the priors fairly tight? Do the posteriors for the inputs (in case they are part of the Bayesian framework) make sense?

[7] Line 373: Different model of parameters for Figure 5 and Figure 4. Is this because the inputs were fitted for Figure 5 but not for Figure 4? Or is it because different parts of the model were trained in Figures 4 and 5.

[8] Related to my previous comment, between Figures 4 and 5, were the same parameters trained in both cases, and if that’s the case, then how different are the posteriors? They are both for rodent data, so one would expect the posteriors to be similar. Is that the case? Cross reference to comparisons of the posteriors is needed.

[9] Line 418: How are the positive and negative inputs prescribed in the model? I understand more details are available in the Methods, but a simple clarification in the results would be helpful. For example, if N_NO neurons are activated in the model to simulate one of the cases then it can be stated without going into all the details of the simulation setup. It is difficult to make sense of the outputs shown in figures without having a very brief reminder of what the input is for the simulation.

[10] Line 422: why are the number of parameters 52 for the macaque data in Figure 6?

[11] Going back to comment [1], so far only Figure 5J has validation data, all other plots seem to be results from model training. This is fine, but then the notion of ‘validation’ data is misleading. What do the authors mean by ‘validation’?

[12] Figure 7: How many parameters were trained for the human case? In previous Figures 4,5,6 there was a different number of parameters per figure. Now I’m curious how many parameters are there for Figure 7. Also, I’m still curious about the posteriors, and whether posteriors from different re-training align with each other.

[12] Line 492: When the authors say that they translated all mechanisms from rodent and macaque to the human case do they mean that after re-training independently to each data set, from rodents, to macaque, to human, the features of the inferred dynamics for the neural responses N_NO, N_Pyr, N_NPY align in all cases? Or was there actually something that the authors did to ‘transfer’ parameter information from one training to another? For instance, did the authors use the posteriors from training in Figure 4 as priors for Figure 5? If every training is done independently, then I’m not sure ‘translation’ is the most accurate word. Even in that case, the fact that the inferred dynamics for N_NO, N_Pyr, N_NPY and their contribution to vasodilation/vasoconstriction is preserved with the different data-sets is remarkable and very interesting.

[13] Eq. 6a shows a very standard eq. for NO from which I don’t expect a negative NO value. Why are there negative NO values in the results?

[14] There are cross references to Section 7, I assume the authors refer to the now-Supplement.

[15] Regarding a previous comment on prior and posterior reporting and comparison between posteriors. The Supplement shows the different posteriors. How close are the posteriors between independent trainings from Figures 4-7 should definitely be done. Also, a table with the priors is needed.

**Have the authors made all data and (if applicable) computational code underlying the findings in their manuscript fully available?**

Reviewer #1: Yes

Reviewer #2: Yes

PLOS authors have the option to publish the peer review history of their article (what does this mean?). If published, this will include your full peer review and any attached files.

Reviewer #1: No

Reviewer #2: No
---

## [Decision Letter · Decision Letter 1]

13 Dec 2022

Dear Dr. Cedersund,

We are pleased to inform you that your manuscript 'A quantitative model for human neurovascular coupling with translated mechanisms from animals' has been provisionally accepted for publication in PLOS Computational Biology.

Best regards,

Alison L. Marsden

Academic Editor

PLOS Computational Biology

Jason Haugh

Section Editor

PLOS Computational Biology

Please make the requested changes the reviewer has asked for and submit the final version.

Reviewer's Responses to Questions

**Comments to the Authors:**

Reviewer #1: The authors have addressed my concerns. I would just make two final comments - first, the response to my first question was very thorough, but did not make it into the manuscript. I believe it would strengthen the paper if included. Secondly, the new table detailing the parameters of each model is very helpful but could be designed better. For instance, by making the x's black or leaving them empty to show that they're not used in the given model, and maybe combining or merging cells to show shared values (for instance, the top three rows could be merged in each of the columns, which would be easier to read than checking to see that every box has a y in it) This would also help you avoid the 1 and 2 superscripts.

Reviewer #2: The authors have replied to each of my comments in a thorough manner, I appreciate it. I don't have other comments.

**Have the authors made all data and (if applicable) computational code underlying the findings in their manuscript fully available?**

Reviewer #1: Yes

Reviewer #2: Yes

PLOS authors have the option to publish the peer review history of their article (what does this mean?). If published, this will include your full peer review and any attached files.

Reviewer #1: No

Reviewer #2: No

---

## [Editor Report · Acceptance letter]

21 Dec 2022

PCOMPBIOL-D-22-00855R1 

A quantitative model for human neurovascular coupling with translated mechanisms from animals

Dear Dr Cedersund,

I am pleased to inform you that your manuscript has been formally accepted for publication in PLOS Computational Biology. Your manuscript is now with our production department and you will be notified of the publication date in due course.

With kind regards,

Anita Estes
